# Regulatory and evolutionary adaptation of yeast to acute lethal ethanol stress

Jamie Yang[1,2], Saeed Tavazoie[1,2,3]*

1 Department of Systems Biology, Columbia University, New York City, New York, United States of America,
2 Department of Biochemistry and Molecular Biology, Columbia University, New York City, New York, United
States of America, 3 Department of Biological Sciences, Columbia University, New York City, New York,
United States of America

* st2744@columbia.edu

## Abstract

The yeast *Saccharomyces cerevisiae* has been the subject of many studies aimed at understanding mechanisms of adaptation to environmental stresses. Most of these studies have focused on adaptation to sub-lethal stresses, upon which a stereotypic transcriptional program called the environmental stress response (ESR) is activated. However, the genetic and regulatory factors that underlie the adaptation and survival of yeast cells to stresses that cross the lethality threshold have not been systematically studied. Here, we utilized a combination of gene expression profiling, deletion-library fitness profiling, and experimental evolution to systematically explore adaptation of *S. cerevisiae* to acute exposure to threshold lethal ethanol concentrations—a stress with important biotechnological implications. We found that yeast cells activate a rapid transcriptional reprogramming process that is likely adaptive in terms of post-stress survival. We also utilized repeated cycles of lethal ethanol exposure to evolve yeast strains with substantially higher ethanol tolerance and survival. Importantly, these strains displayed bulk growth-rates that were indistinguishable from the parental wild-type strain. Remarkably, these hyper-ethanol tolerant strains had reprogrammed their pre-stress gene expression states to match the likely adaptive post-stress response in the wild-type strain. Our studies reveal critical determinants of yeast survival to lethal ethanol stress and highlight potentially general principles that may underlie evolutionary adaptation to lethal stresses in general.

## Introduction

Stress studies in yeast have mostly focused on acquired stress tolerance, the phenomenon in which exposure to a non-lethal dose of stress allows yeast to survive a subsequent exposure to a lethal dose. Exposure to this mild dose of stress activates a general stress response that allows the cells to better survive exposure to other lethal stresses, known as cross protection [1–3]. Many stresses activate the environmental stress response (ESR), which roughly encompasses 900 genes that change their expression in response to the stress [4, 5]. This common response to stress includes about 300 induced genes that are generally involved in stress defense, including genes in the categories of DNA damage repair, cell wall modification, vacuolar functions,

**Data Availability Statement:** The authors submitted their datasets to GEO with accession number: GSE151786.

**Funding:** NIH (R01-AI077562) and NIH (R01-HG009065) to S.T. NIH MSTP award to J.Y. The funders had no role in study design, data collection and analysis, decision to publish, or preparation of

the manuscript S.T. received salary support base on his effort from both NIH grants.

**Competing interests:** The authors have declared that no competing interests exist.

detoxification of reactive oxygen species, and heat shock proteins. The ESR also encompasses 600 repressed genes that encode ribosomal proteins, ribosome biogenesis factors, translation initiation proteins, tRNA synthesis proteins, and proteins involved in other growth-related processes [4, 6]. The expression of these upregulated and downregulated group of genes show identical but opposite patterns. Additionally, the magnitude of the expression changes is proportional to the degree of stress imposed on the cells. Because of the dominance of the ESR in the literature of yeast stress response, we make multiple comparisons to it below.

Many studies have found that the ESR is tightly coupled to growth rate [7–9], and it is prohibitively difficult to optimize the growth and stress defense independently of each other [10]. Studies that utilized the pooled yeast knockout library have found an inverse correlation between growth rate and stress survival, with ribosome biogenesis, translation, and mitochondrial functions providing the link between growth rate and stress defense [8]. Stress-specific pathways are revealed once the dominant growth-rate effects are factored out. Our experiments are set up in such a way as to exclude strains that have reduced growth rate, since this would be a trivial solution to stress defense.

Some studies that investigated acute exposure to lethal stress have suggested that a fraction of cells in a population can survive lethal stress due to either a bet-hedging mechanism in which the sub-population exist in a slow growing protected state [9] or a mild environmental stress response activation through random double-strand breaks [11]. Either of these two scenarios would occur in a small percentage of cells, causing that subset of the population to be slow growing and allowing them to survive acute lethal doses of stress.

In order to better understand the mechanisms of survival in the face of acute lethal stress, we exposed yeast cells to two minutes of lethal ethanol stress, a time period short enough to minimize the activation of any programmed cellular response during the period of stress. We used RNA-seq to discover global transcriptional responses of yeast cells following the acute stress. In addition, we used fitness profiling of the pooled yeast knockout library to determine genes and pathways that contributed to survival under lethal ethanol stress. In order to determine whether and to what extent the survival-rate could evolve, we exposed multiple populations of yeast cells to repeated lethal ethanol stress. This experimental evolution paradigm led to a strain of yeast that showed an order of magnitude improvement in survival without compromising growth rate. Significant differences in global transcriptional responses of the evolved strain accompany its superior survival.

## Results

### Survival of yeast cells following acute lethal exposure to ethanol

We developed an acute lethal stress paradigm in which haploid yeast cells were treated with a brief two-minute exposure to a range of ethanol concentrations (19% - 26%). This brief exposure reduced the possibility of a direct transcriptional response during the period of stress. In this way, we created conditions in which the cellular state immediately prior to the stress and the longer-term transcriptional responses following the short period of stress were dominant contributors to survival. As can be seen from the stress-survival curve (Fig 1A), ethanol exposure above 20% causes lethality as determined by fraction survival of colony forming units. Furthermore, survival drops exponentially for concentrations within the lethal range, reaching $10^{-5}$ at 26% ethanol.

### Global transcriptional response at the threshold of lethal ethanol stress

In order to gain better insights into the cellular pathways that may contribute to acute ethanol stress survival, we carried out transcriptional profiling using RNA-seq [12]. To minimize any

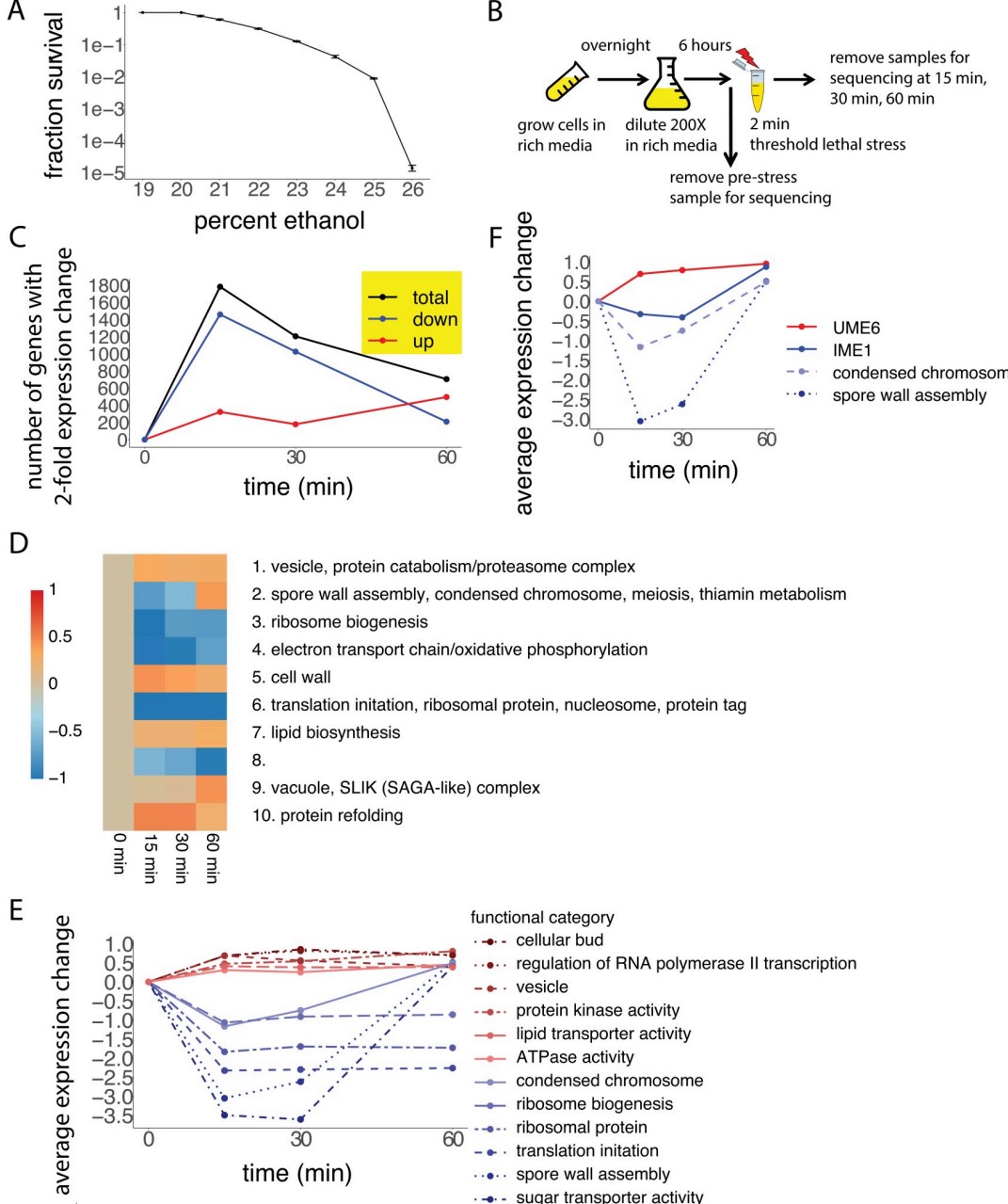

**Fig 1. Global transcriptional response to threshold lethal acute ethanol stress.** (A) The fraction survival of yeast CFUs exposed to two minutes of ethanol stress, from 19% to 26% ethanol. Error bars represent standard error. (B) Procedure used to perform stress assay on yeast cells and to collect samples for RNA-seq. (C) Number of differentially expressed genes, for each post-stress time point relative to the pre-stress time point, measured as the number of genes with more than a two-fold expression change. The total number of differentially expressed genes was broken down into those with a two-fold downregulation and those with a two-fold upregulation. (D) Clustering of the time-course gene expression data using k-means with 10 clusters. Gene ontology analysis was performed on each of the clusters, and significant functional categories (p<0.001) are shown. (E) Average expression from the 12 most significant gene ontology categories in the 15- or 30-minute post-stress time point. Red colored lines indicate upregulated genes and blue colored lines indicate downregulated genes within the early post-stress time points. (F) Average expression of condensed chromosome and spore wall assembly genes, along with the expression of UME6 and IME1.

confounding effects from dying cells, we chose to determine transcriptional responses at the threshold level of lethality corresponding to 20% ethanol. The transcriptional state of the haploid yeasts (strain BY4741) were monitored starting prior to stress exposure, and followed until one hour after stress exposure, sampling at various time points during the recovery from stress (Fig 1B).

The samples collected from the experiment were analyzed by RNA-seq. Comparison of global transcriptional states between each post-stress time point and the pre-stress time point revealed transcriptional changes that accompany any adaptation following stress. Fig 1C shows the number of genes that exhibit a two-fold change in expression at each post-stress time point compared to the pre-stress time point. The number of genes that showed a two-fold change is greatest at the 15-minute time point, suggesting that the peak response to ethanol stress occurs early. At the early time points (15 and 30 minutes), there are roughly five times as many genes (~1500) with a two-fold decrease than a two-fold increase (~300), indicating that the transcriptional response to ethanol stress is largely driven by a global downregulation of gene expression.

To discover dominant patterns of gene expression following stress, we carried out unsupervised clustering of genes using the k-means algorithm [13]. The gene expression patterns of the ten resulting clusters are shown in Fig 1D, demonstrating both induced and repressed groups of genes. The application of iPAGE [14], a pathway discovery algorithm, revealed that these expression clusters were significantly enriched in various functional categories and biological processes (Fig 1D). There are 4 clusters (3, 4, 6, 8) that show decreased expression over the entire recovery period. Ribosome biogenesis, ribosomal proteins, and translation are three well-documented functional categories, part of the environmental stress response, that decrease in expression in response to a variety of stresses [4, 5]. Cluster 10 shows the strongest upregulated response during the early time periods, suggesting the possible adaptive role of protein refolding by heat shock proteins in the adaptation to ethanol stress. This is consistent with previous studies that have shown induction of heat shock proteins in response to mild ethanol stress (4–10% ethanol) [15]. Cluster 7 contains upregulated genes in lipid biosynthesis, more specifically, sphingolipid biosynthesis. Membrane lipid composition is known to play an important role in sublethal ethanol and heat stress [16], and increased sphingolipid biosynthesis has been shown to increase tolerance to heat stress in yeast [17]. Other studies have shown that vesicular and vacuolar transport are both important in yeast survival to mild ethanol stress [18–23], represented here in clusters 1 and 9, respectively. While induction of vesicular genes occurs shortly after exposure to ethanol, upregulation of vacuolar functions shows a delayed induction, peaking closer to 60 minutes after the onset of ethanol stress.

In addition to partitional clustering using the k-means algorithm, we also performed hierarchical clustering. The resulting dendrogram is shown in S1 Fig. Clustering by both genes and time points, we see that the 15-minute and 30-minute post-stress time points are most similar, with the pre-stress time point differing the most from all the post-stress time points.

In order to capture the broadest possible set of pathways modulated during the adaptation process, we compared the gene expression at each of the post-stress time points to the pre-stress time point and discovered gene ontology classes that are significantly enriched in them. The complete list of these categories can be found in S1 Table. The average gene expression from the 12 most significant functional categories modulated during the early time points (15 and 30 minutes) are shown in Fig 1E. Whereas lipid biosynthesis was significant in one of the upregulated patterns in Fig 1D, here lipid transporter is also shown to be significantly induced, further suggesting its importance in ethanol stress response. Regulation of RNA polymerase II transcription [24] and protein kinase activity [25] are both categories that are known to be upregulated in response to a variety of stresses. The ATPase activity category consists of some

genes that code for heat shock proteins, including the Hsp70 family, all of which have been shown to protect against mild ethanol stress [15]. The ATPase activity category is also composed of genes that code for vacuolar H⁺-ATPase, required for tolerance to straight chain alcohols through vacuolar acidification [22]. The category that is upregulated the greatest during the early time points, cellular bud, is composed of genes involved in creating a protuberance from the mother cell to create a daughter cell, which have not been previously implicated in ethanol stress adaptation. This group of genes have functions in polarized growth, localization of proteins to either mother or daughter cell, cytokinesis, and cell wall formation [26].

Some of the processes upregulated from Fig 1D and 1E are related to damages that are known to be caused by ethanol. The dominant process by which yeast cells die from high ethanol concentrations is increased membrane fluidity and subsequent decrease in membrane integrity [27]. Other biological functions are also damaged upon ethanol exposure [28], which are summarized in Table 1, along with how the cells in our experiments responded to ethanol stress in that category. The upregulation of processes that are normally disrupted by ethanol suggests that our post-stress cells are mounting a response to counteract the damaging effects of ethanol.

The functional category that shows the greatest downregulation is sugar transporter activity. Other transport processes, such as ammonium transport and organic acid transport, are also downregulated in response to ethanol stress in our data, but to a lesser extent. Transport processes play an important role in the ethanol stress response, with some reports of general inhibition of transport processes [29] as well as reports of an increase of transport processes, specifically hexose transport [30] under mild ethanol stress. Deletion of transport genes has also been shown to hurt survival when yeast cells were exposed to mild ethanol stress [20]. Vesicle-mediated transport was upregulated in our dataset, suggesting both an increase and decrease of transport processes at threshold lethal ethanol stress, depending on the substrate being transported.

Two categories, spore wall assembly and condensed chromosome, show an initial decrease during the early recovery period with a subsequent increase in gene expression at 60 minutes of recovery (Fig 1D, cluster 2. Average expression change shown in Fig 1E). Chromosome condensation occurs during meiosis in yeast [31]. Meiosis and spore formation are processes that are specific to diploid yeast [32], and their surprising modulation in haploid yeast suggests their potentially novel adaptive value in response to acute lethal ethanol stress. The application of FIRE [33], a *cis*-regulatory motif discovery algorithm, on cluster 2 of Fig 1D revealed two overrepresented motifs (S2 Fig). One of these was the URS1 motif (5′-GGCGGC-3′), which is recognized by UME6/IME1, a major transcriptional regulator of early meiosis genes [34, 35]. While UME6 directly binds the URS1 motif and is a key transcriptional repressor of early meiosis genes, IME1 activates transcription of these genes through its interaction with UME6. Fig 1F shows the expression changes of *UME6* and *IME1* in relation to the average expression changes of spore wall assembly and condensed chromosome genes. The initial decrease in spore wall assembly and condensed chromosome expression could be explained by an upregulation of the repressor UME6 and downregulation of the activator IME1, but by 60 minutes post-stress, the activating effects of IME1 dominate.

**Table 1. Known influence of ethanol on yeast cells and its corresponding post-stress response in our dataset.**

| Ethanol influence | Response to threshold lethal ethanol stress |
|---|---|
| increased membrane fluidity | upregulation of lipid biosynthesis |
| inhibition of H⁺-ATPase | upregulation of H⁺-ATPases |
| inhibition of transport processes | upregulation of lipid transport |
| disruption of vacuole morphology | upregulation of vacuolar functions |

## Contribution of all non-essential genes to survival under acute exposure to lethal ethanol stress

Transcriptional responses to extreme stress can reflect pathways that may be adaptive. Alternatively, these gene expression dynamics may reflect nonadaptive or even maladaptive responses that may only be of value within the organism's native habitat [36]. In order to systematically determine the contribution of yeast genes to acute lethal stress, we carried out, in triplicate, fitness profiling of a pooled haploid yeast deletion library [37] upon exposure to lethal ethanol stress (Fig 2A). We used 24.5% ethanol, which is the concentration at which 1% of the wild-type population survives after a 2-minute exposure. The inclusion of a post-stress outgrowth phase reduced the likelihood that some mutants may achieve better survival due to a severe growth defect. This haploid library contains 4,500 deletions of most non-essential genes, with each gene deletion represented by a unique 20-nucleotide barcode [37]. Comparing the abundance of post-stress gene deletions to pre-stress gene deletions, we were able to systematically quantify the effects of each gene deletion on survival.

This analysis revealed gene deletions that significantly increased or decreased survival. There were 192 gene deletions that significantly improved survival and 735 gene deletions that significantly diminished survival (see Materials and Methods). The list of the top gene

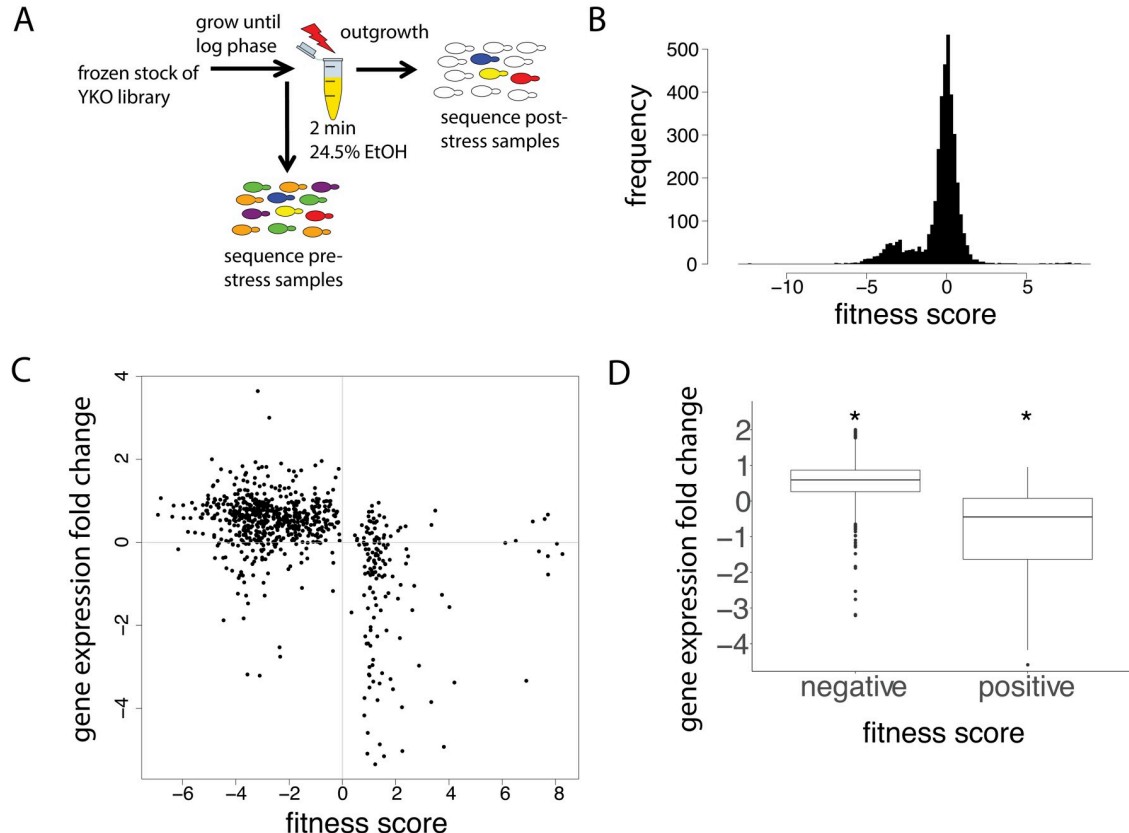

**Fig 2. Utilizing the pooled yeast deletion library to determine contributions of all non-essential genes to survival.** (A) Procedure used to perform stress assay on yeast cells and to determine contribution of each gene deletion to survival. (B) Histogram of fitness (survival) scores. (C) Maximum gene expression fold change versus statistically significant fitness scores to determine concordance between a gene's transcriptional response and its contribution to stress survival. (D) Boxplot showing that negative fitness scores have a significant positive expression change ($p < 2.2 \times 10^{-16}$), and positive fitness scores have a significant negative expression change ($p = 1.9 \times 10^{-13}$).

**Table 2. Top 20 most enriched and depleted gene deletions upon exposure to lethal ethanol stress based on fitness score.**

**ENRICHED**

| Gene Name | Systematic Name | Score | Description |
|---|---|---|---|
| AAH1 | YNL141W | 8.26 | adenine deaminase |
| LTV1 | YKL143W | 8.05 | EGO/GSE complex subunit, upstream of TOR complex |
| FTR1 | YER145C | 7.72 | iron permease |
| RPL13A | YDL082W | 7.72 | ribosomal 60S subunit |
| SNT1 | YCR033W | 7.71 | deacetylase, positive regulation of stress-activated MAPK cascade |
| AFG3 | YER017C | 7.59 | mitochondrial metallopeptidase |
| RPS16A | YMR143W | 7.47 | ribosomal 40S subunit |
| TRM13 | YOL125W | 7.38 | Methyltransferase |
| EAP1 | YKL204W | 7.15 | translation initiation factor, TOR pathway |
| RPL14A | YKL006W | 6.90 | ribosomal 60S subunit |
| RPL16B | YNL069C | 6.80 | ribosomal 60S subunit |
| PIL1 | YGR086C | 6.51 | eisosome assembly |
| KSS1 | YGR040W | 6.11 | MAP kinase |
| PET117 | YER058W | 5.86 | electron transport chain |
| ASF1 | YJL115W | 4.20 | nucleosome assembly factor, stress response |
| MRP7 | YNL005C | 4.02 | mitochondrial ribosomal large subunit |
| VPS24 | YKL041W | 3.81 | vacuolar protein sorting |
| CTR1 | YPR124W | 3.74 | copper transporter |
| REG1 | YDR028C | 3.49 | protein phosphatase regulator |
| GAT2 | YMR136W | 3.34 | transcription factor |

**DEPLETED**

| Gene Name | Systematic Name | Score | Description |
|---|---|---|---|
| VPS69 | YPR087W | -12.4 | vacuolar protein sorting |
| SIW14 | YNL032W | -6.92 | tyrosine phosphatase |
| NGG1 | YDR176W | -6.81 | Acetyltransferase |
| YJL047C-A | YJL047C-A | -6.79 | unknown function |
| DGA1 | YOR245C | -6.38 | acyltransferase, triglyceride biosynthesis |
| INO2 | YDR123C | -6.34 | transcription factor, phospholipid biosynthesis |
| IRC23 | YOR044W | -6.16 | unknown function |
| VPS13 | YLL040C | -5.97 | vacuolar protein sorting |
| SUE1 | YPR151C | -5.96 | degradation of cytochrome c |
| UGA4 | YDL210W | -5.79 | GABA transport, located on vacuole membrane |
| MON1 | YGL124C | -5.70 | guanine nucleotide exchange factor, located on vacuole membrane |
| DUR3 | YHL016C | -5.64 | putrescine, spermidine, urea transporter |
| ISN1 | YOR155C | -5.53 | nucleotidase, breaks IMP to inosine |
| AHK1 | YDL073W | -5.52 | scaffold protein, important in osmotic stress |
| YCF1 | YDR135C | -5.32 | glutathione transporter, located on vacuole membrane |
| PMC1 | YGL006W | -5.28 | ATPase, located on vacuole membrane |
| YGL015C | YGL015C | -5.25 | unknown function |
| YGL140C | YGL140C | -5.24 | unknown function |
| ICL1 | YER065C | -5.24 | isocitrate lyase, induced by growth on ethanol |
| TUF1 | YOR187W | -5.24 | mitochondrial translation elongation factor |

deletions with positive and negative fitness effects are shown in Table 2. For the gene deletions that diminished survival, six out of the top twenty have vacuolar functions, suggesting their essentiality for yeast survival under lethal ethanol stress. For the gene deletions that improved survival, five out of the top twenty are non-essential ribosomal subunits [38], three are

mitochondrial proteins [39–41], and two are in the TOR (target of rapamycin) pathway [42, 43]. Increased survival from deletion of ribosomal and mitochondrial genes are consistent with prior yeast deletion studies showing a similar benefit under various other stresses [8, 44]. Decreased expression of TOR pathway components is well known to increase survival under a broad range of stresses [45–47]. More ribosomal genes are revealed if we expand the list to the top 100 gene deletions. In fact, when we performed iPAGE analysis on the full range of scores, we found significant enrichment for ribosome genes at the positive end of the fitness distribution, suggesting that ablation of non-essential components of translation are beneficial under acute lethal stress (S3 Fig).

S2 Table summarizes the pathways and genes that are affected during mild ethanol stress exposure and compares the directionality of their post-stress responses to the response at threshold of lethality determined here. From the top section of the table, it is clear that all pathways affected by mild stress are also affected in the same direction in both our transcriptional profiling and deletion library experiments. At the gene level, however, only 20 out of 50 genes trend in the same direction from our RNA-seq data, and only 8 out of 50 genes agree from our deletion library data. Individual genes do not overlap significantly between mild and lethal ethanol stress. In fact, even between 8% and 11% ethanol, different genes were associated with ethanol tolerance [28]. However, on the pathway level, there is significant agreement between the responses to mild and lethal ethanol stress.

To determine whether the transcriptional reprograming was adaptive, we asked whether there was a significant association between the fitness effect of a gene deletion and the maximal transcriptional change of a gene across the post-stress time course. Plotting all significant positive and negative fitness scores of gene deletions against their maximal post-stress expression fold-change, we noticed that ~85% (639 of 754) of all significant genes exhibited a negative relationship (Fig 2C and 2D), suggesting that much of the global transcriptional reprograming likely plays an adaptive role under acute lethal ethanol stress.

## Experimental evolution of increased survival upon lethal ethanol stress

The genetic perturbations in the yeast deletion library enabled us to determine the extent to which loss-of-function mutations substantially improves survival of yeast cells to lethal ethanol stress. We were curious to what extent random mutations and selection may drive improved lethal stress survival and whether gene expression reprogramming may be a key element in the improved survival. To this end, we developed a laboratory experimental evolution paradigm in which wild-type haploid yeast cells were exposed to lethal ethanol stress and then recovered in an outgrowth phase over multiple cycles of evolution. In each round, cells were first grown to saturation in rich media, diluted and grown until mid-log phase, then exposed to two minutes of lethal ethanol stress. The surviving cells were then regrown to saturation, thus starting the next round of selection (Fig 3A). Whereas some studies use nutrient-limited media during the regrowth phase so as to minimize growth-based competition [48], we wished to avoid accumulation of mutations that cause generic slow-growth, which would be a trivial solution to lethal stress survival. Our strategy favored mutations that increased survival without compromising bulk growth rate. As such, we decided to recover the cells in rich media following lethal stress exposure. Since slow-growing beneficial mutations will be positively selected for during stress exposure and negatively selected for during recovery, it may explain the non-monotonic and cyclical pattern of survival throughout the laboratory evolution process (S4 Fig).

Experimental evolution was carried out in triplicate populations. The lethal selection was calibrated to have a baseline survival of ~1%. The best performing evolved population showed an average of ~35% survival compared to the parental strain (Fig 3B, red vs. black). Individual

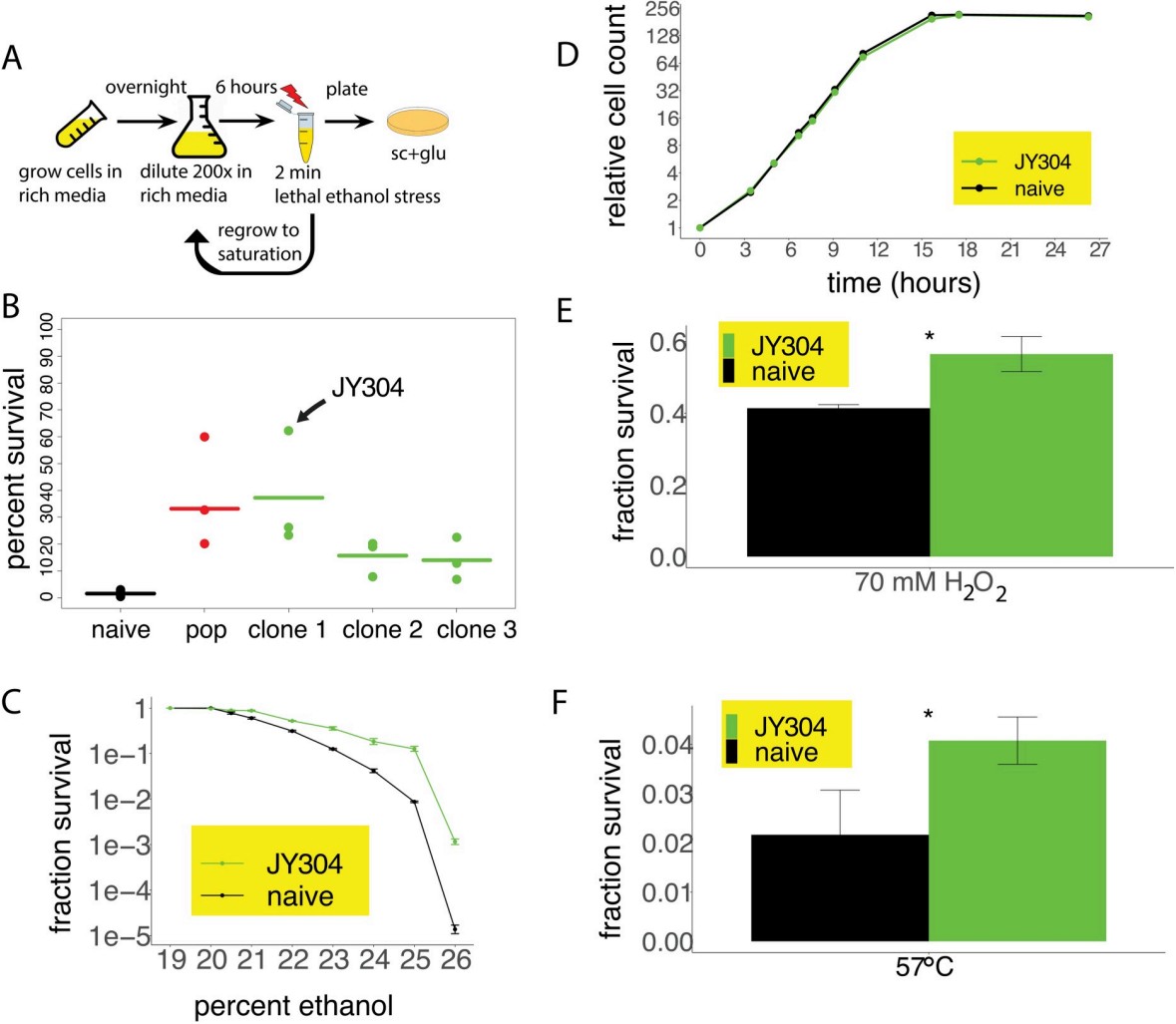

**Fig 3. Laboratory experimental evolution to repeated acute lethal ethanol stress selects for mutants with substantially enhanced survival.** (A) Laboratory evolution protocol. At every round of selection, survival was measured by plating, and experiments were stopped when survival was enhanced substantially beyond the baseline level of 1%. (B) Fraction survival of the naive wild-type strain (black), evolved population (red), and 3 distinct clones from the evolved population (green). Horizontal lines indicate the average survival of each of the replicates. JY304 is derived from the colony with the highest percent survival. (C) Analogous to Fig 1A, comparing the fraction survival of strain JY304 to the wild-type strain at a range of ethanol concentrations, from 19% to 26%. Error bars indicate standard error. (D) Growth curves of strain JY304 and wild-type strain. (E) Testing cross-protection of cells evolved under ethanol stress, compared to the wild-type strain, stressed at 70 mM hydrogen peroxide. Error bars indicate standard error. (F) Same as D except with heat stress at 57°C.

colonies exhibited an average survival of at least 10% (Fig 3B, green). The colony with the highest average fraction survival (hereinafter referred to as strain JY304) was chosen for subsequent phenotypic and molecular analyses. We first determined the survival advantage of strain JY304 across a range of ethanol concentrations (Fig 3C). At the highest concentration tested (26%), we saw nearly a hundred-fold increase in survival. Remarkably, this survival advantage was not accompanied by any significant reduction in the bulk growth rate of the population (Fig 3D).

## Cross-resistance of the evolved strain across orthogonal lethal stresses

The survival advantage of the strain JY304 may be due exclusively to ethanol-specific advantages conferred by the underlying mutations. Alternatively, at least some of the survival

advantage may be due to more general survival effects beyond lethal ethanol stress. To see any evidence for such cross-resistance, we determined haploid yeast survival to orthogonal stresses: 70 mM hydrogen peroxide and 57°C heat stress, both for a duration of two minutes. As can be seen in Fig 3E and 3F, strain JY304 showed significantly higher survival for both stresses, suggesting that at least part of the survival advantage is non-specific to ethanol.

## Evolution of transcriptional responses in the evolved hyper-surviving strain

The evolved strain JY304 may achieve superior ethanol survival due to an improved ability to reprogram gene expression during the minutes and hours following exposure to lethal stress. Alternatively, some of the survival advantage may be due to establishment of a pre-stress cellular state that is more resistant to lethal stress. In order to determine the extent to which these differing strategies may contribute to survival, we carried out transcriptional profiling of strain JY304 exactly as performed for the parental wild-type strain. The parental strain showed a dramatic shift in gene expression in the fifteen minutes post-stress, repressing and inducing a large number of genes that led to a substantial divergence in the global transcriptional state, followed by a gradual recovery of gene expression back towards the pre-stress patterns over an hour (Fig 1C). The evolved strain, however, exhibited a less dramatic shift in gene expression that was largely maintained in the post-stress period without significant recovery (Fig 4A).

To discover significant patterns of gene expression, we performed unsupervised clustering using the k-means algorithm with k = 10. This revealed gene expression clusters, six of which showed the same enrichment in functional categories as observed in the parental strain. Three of these categories were upregulated (vesicle, proteasome complex, protein refolding) and three were downregulated (ribosomal protein, ribosome biogenesis, nucleosome). Although the directionality of these gene expression dynamics was similar to the parental strain, the magnitude of change was significantly different. All six categories showed a lower magnitude of change in strain JY304 compared to the wild-type strain. The overall lower magnitude of change is evident in the comparison of the heat maps (Fig 4B).

In addition to performing k-means clustering, we also carried out hierarchical clustering. The resulting dendrogram is shown in S5 Fig. Clustering by both genes and time points, we see that the 15-minute and 30-minute post-stress time points are most similar, with the pre-stress time point differing the most from all the post-stress time points. This pattern is the same as that of the wild-type strain.

To take a closer look at the pre-stress state of the evolved and parental strains, we compared the global expression of genes in strain JY304 to the wild-type strain during the exponential growth phase in rich media in the absence of stress. Remarkably, we saw significant differences in pre-stress gene expression states: 266 genes were higher in expression more than two-fold and 52 genes were lower in expression more than two-fold between strain JY304 and the wild-type strain. Table 3 contains a list of the twenty most highly differentially expressed genes. Many of the most highly differentially expressed genes are of unknown function, suggesting the potential importance of novel genes and pathways that have yet to be elucidated, but may play an important role in the ethanol stress response. There are also several categories of genes that are seen in both the increased and decreased expression categories: transposable element, heat shock protein, helicase, and mitochondrial protein.

To explore the difference between the pre-stress state of strain JY304 and the parental strain at the level of significantly enriched pathways, we performed iPAGE analysis, which revealed 53 significant gene ontology terms with significant differences in expression (p<0.001), clearly showing a difference in the pre-stress cellular state between these two strains (S3 Table).

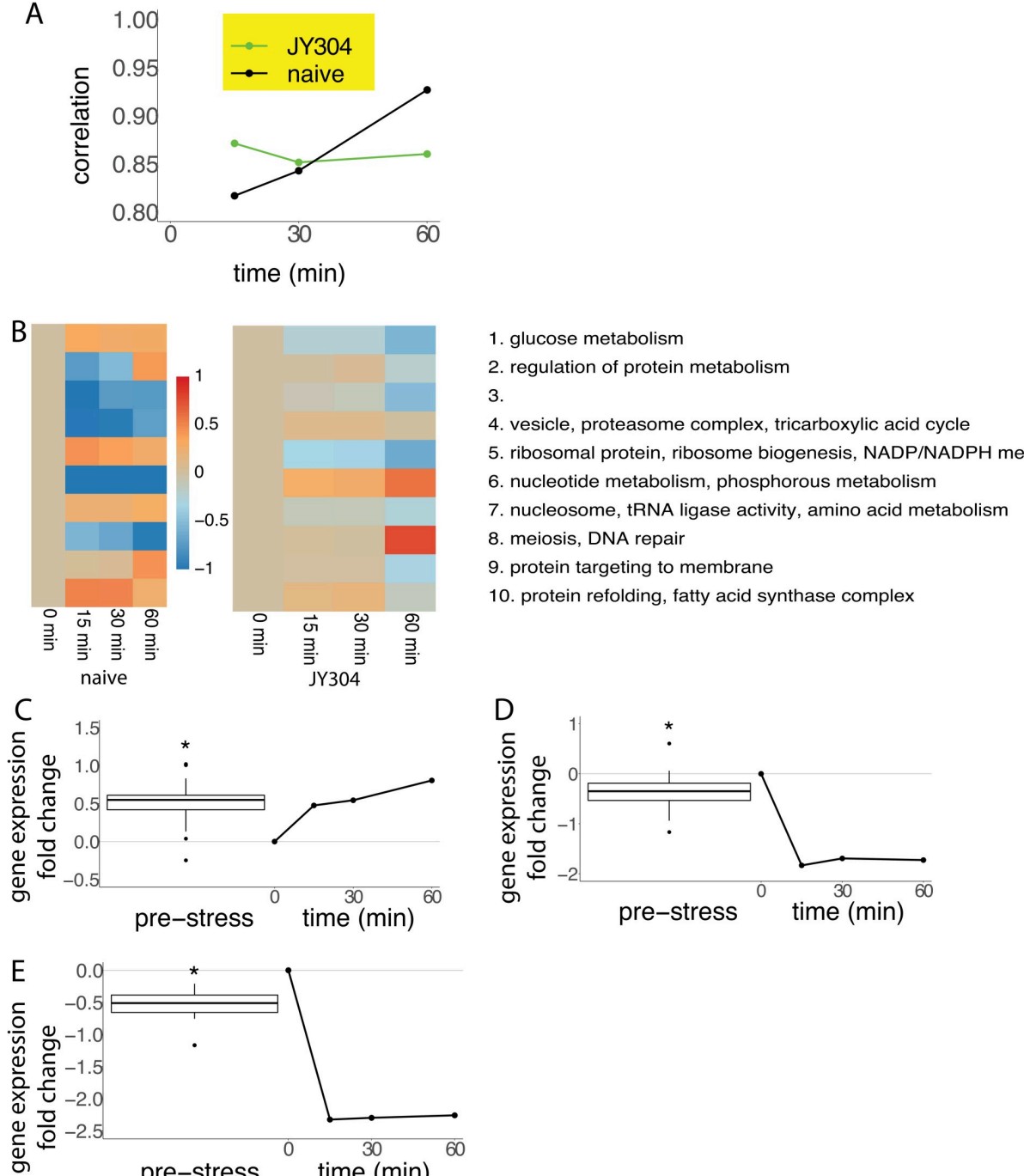

**Fig 4. Transcriptional responses of an extreme ethanol stress survivor.** (A) Correlation between each of the post-stress time points to the pre-stress time point for both strain JY304 and the wild-type strain. (B) Clustering of the time-course gene expression data using k-means with 10 clusters for strain JY304. Gene ontology analysis was performed on each of the clusters, and significant functional categories (p<0.001) are shown for each cluster. For comparison, the heatmap for the wild-type time-course gene expression clustering is shown to the left. Note: The significant functional categories for strain JY304 do not apply to the wild-type heatmap, which is unaltered from Fig 1D. (C-E) Average expression of the three functional categories that are significantly (p<0.001) up- or downregulated in the wild-type strain in response to ethanol stress (right), along with a boxplot of their corresponding pre-stress expression in strain JY304. The categories are kinase (C) ($p < 2.2 \times 10^{-16}$), ribosomal protein (D) (p = 0.005415), and translation (E) (p = 0.00324).

**Table 3. Top 20 most differentially expressed genes in both the positive and negative directions in strain JY304 compared to the parental strain during exponential growth.**

### INCREASED EXPRESSION

| Gene Name | Systematic Name | Fold increase | Description |
|---|---|---|---|
| YMR046C | YMR046C | 26.7 | transposable element |
| YKL131W | YKL131W | 6.7 | unknown function |
| YLR410W-A | YLR410W-A | 6 | transposable element |
| HSP32 | YPL280W | 5.5 | heat shock protein |
| YBL111C | YBL111C | 4.9 | helicase-like protein |
| YHL037C | YHL037C | 4.8 | unknown function |
| YHL005C | YHL005C | 4.8 | unknown function |
| YGL235W | YGL235W | 4.4 | unknown function |
| ATP6 | Q0085 | 4.3 | mitochondrial protein |
| AI1 | Q0050 | 4.2 | mitochondrial protein |
| YPL060C-A | YPL060C-A | 3.8 | transposable element |
| SDP1 | YIL113W | 3.8 | stress-inducible MAP kinase phosphatase, shifts location upon heat stress |
| YDR261C-C | YDR261C-C | 3.7 | transposable element |
| YPR136C | YPR136C | 3.7 | unknown function |
| YBR224W | YBR224W | 3.6 | unknown function |
| RRT16 | YNL105W | 3.5 | unknown function |
| COX3 | Q0275 | 3.4 | mitochondrial protein |
| AI2 | Q0055 | 3.4 | mitochondrial protein |
| YPR197C | YPR197C | 3.3 | unknown function |
| COB | Q0105 | 3.2 | mitochondrial protein |

### DECREASED EXPRESSION

| Gene Name | Systematic Name | Fold decrease | Description |
|---|---|---|---|
| YGL088W | YGL088W | 75.9 | unknown function |
| YDR545C-A | YDR545C-A | 38.3 | unknown function |
| SNA3 | YJL151C | 9.8 | vesicular sorting of proteins to vacuole |
| YIL177C | YIL177C | 7.9 | helicase |
| SRB6 | YBR253W | 6.7 | RNA polymerase II subunit |
| CUE4 | YML101C | 5.2 | unknown function |
| YGR027W-A | YGR027W-A | 4.9 | transposable element |
| TAR1 | YLR154W-C | 4.7 | mitochondrial protein |
| YLR156W | YLR156W | 3.9 | unknown function |
| COX13 | YGL191W | 3.8 | mitochondrial protein |
| PGA2 | YNL149C | 3.6 | protein transport |
| HSP33 | YOR391C | 3.4 | heat shock protein |
| HHT1 | YBR010W | 2.9 | histone H3, rRNA transcription |
| YBR064W | YBR064W | 2.8 | unknown function |
| COF1 | YLL050C | 2.8 | golgi to plasma membrane transport |
| YAR010C | YAR010C | 2.8 | transposable element |
| YJR028W | YJR028W | 2.8 | transposable element |
| YPR137C-A | YPR137C-A | 2.8 | transposable element |
| RPB9 | YGL070C | 2.7 | RNA polymerase II subunit |
| HTB2 | YBL002W | 2.5 | Histone H2B |

Examination of the three most significant functional categories (kinase, ribosomal protein, and translation) showed that in strain JY304, there was a reprogramming of gene expression such that its pre-stress state was shifted towards that of the wild-type strain at 15 minutes post-

stress (Fig 4C–4E). This suggests that strain JY304 is already in a more stress-resistant state during normal growth and is better prepared to survive an acute transition to lethal stress. In fact, when we compared the pre-stress expression profile of strain JY304 to the post-stress expression profile of the wild-type strain, we noticed a striking Pearson correlation of 0.88 between the two expression profiles. The kinase, ribosomal protein, and translation categories are all part of a general stress response, explaining why strain JY304 also better survives hydrogen peroxide and heat stress (Fig 3E and 3F).

## Discussion

We have examined yeast survival and adaptation to acute exposure to lethal ethanol stress. A novel aspect of our experimental design is that the two-minute ethanol exposure minimizes the time for cells to mount a transcriptional response, making the cell's survival dependent on either its pre-existing state prior to the onset of stress or transcriptional response following the stress. Many of our results, however, are consistent with findings from previous yeast stress experiments. A two-minute exposure to threshold lethal ethanol stress is enough to activate the repressed portion of the ESR, as seen by the significant downregulation of ribosome biogenesis, ribosomal proteins, and translation in our data. We did not, however, see a significant upregulation of genes induced in the ESR. While the ESR shows twice as many genes downregulated (~600) than upregulated (~300) (4), we see five times as many genes downregulated (~1500) than upregulated (~300). Additionally, while the induced and repressed responses of the ESR show equal but opposite magnitude, our repressed response was much stronger, between a two-fold and eleven-fold downregulation on average for significantly repressed functional categories. In contrast, all our most significantly upregulated functional categories were only upregulated less than two-fold, on average (Fig 1E). It would be interesting to explore other stresses at the threshold of lethality to see if they show a similar imbalance between their upregulated and downregulated responses. S6 Fig summarizes the most important functional categories we discovered in response to ethanol stress at the threshold of lethality.

Two gene functional categories, chromosome condensation, which only occurs during meiosis, and spore wall assembly, have only previously been known to occur in diploid yeast. Their significant downregulation shortly after acute ethanol stress suggests a possible adaptive role. In diploid yeast, starvation stress propels yeast cells to undergo meiosis and sporulation [49], but the downregulation of these genes in our data may indicate a novel function. Chu et al., 1998 [50] performed time-course transcriptional profiling during meiosis in yeast to determine all the genes that were significantly induced during meiosis, categorizing them into early, early middle, middle, and late middle stages. The genes from our chromosome condensation and spore wall assembly categories that overlapped with their data were all found to be induced in the early stage of meiosis. Most of the genes known to be involved in the early stage of meiosis function in meiotic prophase, which consists of pairing of homologous chromosomes and recombination. About one-third of these 62 genes contain, in their upstream region, the conserved URS1 motif, which was also found in our analysis. Under heat shock, protein binding at URS1 is lost [51], which allows *HSP82* to be de-repressed and subsequently activate *HSF1* [52]. How this process works in haploid yeast is unknown. Moreover, the role of the URS1 element under ethanol stress has not been previously studied.

It is important to note that the pooled yeast deletion library contains only non-essential gene deletions. It is likely that some essential genes also play an adaptive role in lethal ethanol stress survival, but they are not accessible to examination in the deletion library. A possibly

way to test those genes would be to use temperature-sensitive mutants, a heterozygous diploid library, or CRISPR-interference [53].

When comparing the response to ethanol at 20% (transcriptional profiling of wild-type cells) versus the response at 24.5% ethanol (fitness profiling of deletion library), we are making the assumption that the core response to ethanol stress is similar across the regime spanning 20–24.5% ethanol. As shown by the top section of S2 Table, the pathways affected by mild ethanol stress versus acute lethal ethanol stress are very much in agreement, and we believe this core response extends into the realm of lethality. For example, downregulation of ribosomal proteins and translation are universal responses at various ethanol concentrations below, at, and above 20% ethanol. We do acknowledge, however, that there may be some differences in the transcriptional responses between 20% and 24.5% ethanol. Our analysis therefore attempts to capture the most significant global similarities and differences.

The question of whether the post-stress transcriptional response in our wild-type cells is adaptive is a difficult one to conclusively answer. We believe that it does confer a fitness benefit for a couple of reasons. First, our pooled yeast deletion library experiments showed a significant negative association between transcriptional change and fitness, as shown in Fig 2C and 2D. Second, as shown in Table 1, the post-stress upregulation of lipid biosynthesis, ATPases, lipid transport, and vacuolar functions, all of which are normally disrupted by ethanol, suggests that the cells have mounted a response to counteract the damaging effects of ethanol. However, in order for us to conclusively demonstrate that our post-stress cells are adaptive, additional experiments would need to be performed to test individual genes of interest.

Recent studies suggest that stress defense and growth rate cannot be optimized independently of each other. Zakrzewska et al., 2011 [8], using the haploid pooled yeast deletion library, demonstrated a growth rate of 0.34 doublings per hour corresponding to a 1% survival and a growth rate of 0.26 doublings per hour corresponding to 30% survival, regardless of the type of stress used. Lu et al., 2009 [7], using heat stress, demonstrated a growth rate of 0.44 doublings per hour corresponding to 1% survival and a growth rate of 0.22 doublings per hour corresponding to 30% survival. Both of these studies clearly demonstrated a negative correlation between growth rate and stress survival. Our study, however, showed that our evolved strain JY304, with a survival greater than 30%, had a bulk growth rate comparable to that of the wild-type strain.

While it may seem that we have evolved a strain that is able to optimize both growth rate and stress survival, we do not know the distribution of growth rates in the population. It is possible that strain JY304 exhibits a bet-hedging effect in which a small fraction of the population grows at a slow rate and, thus, is partially protected from acute lethal stress. Although we have clearly shown that part of the increased survival of strain JY304 is due to a bulk gene expression shift to a post-stress state, we are unable to determine whether and how much additional survival benefit is achieved through a slow growing subpopulation since our experiments only measured bulk growth rate.

While the survival advantage of strain JY304 was not accompanied by any significant reduction in the bulk growth rate of the population, there is a possibility that certain environments exist in which strain JY304 would exhibit lower fitness than the wild-type strain. Just like the ESR conveys general stress protection, it does not explain behavior in some laboratory stresses, such as various nutrient deprivation [6]. It is certainly possible that strain JY304 does not perform optimally in those environments as well. Additionally, this lack of tradeoff between growth rate and survival applies to acute lethal stresses, but we have not determined how the two strains compare under stresses of long durations. How our evolved strain holds up in numerous other laboratory conditions and environments would be an important subject for future studies.

We have shown that acute exposure of *S. cerevisiae* to ethanol at levels beyond 20% leads to exponential loss of viability. We have determined that the response of haploid yeast cells at the threshold of lethality is characterized by global reprogramming of gene expression largely matching the ESR response in the downregulated genes but with additional unique features including pathways that operate during meiosis in diploid cells. Parallel survival profiling of the yeast deletion library under lethal ethanol stress suggests that much of the post-stress transcriptional reprogramming is adaptive. We utilized laboratory experimental evolution to determine the extent to which survival to lethal ethanol stress could be enhanced and identified strains that had substantially improved survival across a range of ethanol concentrations beyond 20%. Contrary to expectation, we found that enhanced survival was not accompanied by a reduction in bulk growth-rate. Transcriptional profiling of one such hyper-resistant strain showed that, at least, part of the survival advantage can be attributed to regulatory rewiring that establishes a pre-adapted transcriptional state in non-ethanol stressed cells. Our studies provide novel insights into the mechanisms of adaptation to acute lethal stress and the strategies by which cells may improve survival.

## Materials and methods

### Media and growth conditions

We used YPD broth (10 g/L Bacto yeast extract, 20 g/L Bacto peptone, 20 g/L glucose) and YPD agar plates (YPD broth with 20 g/L Bacto agar) for routine growth of yeast strains. All experiments were performed using standard complete + glucose (sc+glu) media (6.7 g/L yeast nitrogen base (Difco), 2 g/L yeast synthetic drop-out mix (US Biologicals), 20 g/L glucose) and sc+glu plates (sc+glu media with 20 g/L Bacto agar). All growth on plates occurred at 30°C. All growth in liquid media also occurred at 30°C and shaking at 220 rpm in an Innova 42 incubator (New Brunswick). To test growth rate, the number of cells were tracked by measuring optical density at a wavelength of 660 nm using an Ultrospec 3100 pro spectrophotometer (Biochrom).

### Yeast strains

All yeast strains were derived from BY4741 (Mat **a** his3Δ1 leu2Δ0 met15Δ0 ura3Δ0) [54], which is the strain we refer to as wild-type. All deletion strains were obtained from the *S. cerevisiae* knockout collection [55].

### Stress experiments

All stress experiments, regardless of background or using the pooled yeast deletion library, were performed identically. Cells were first grown to mid-log phase in 25 mL sc+glu in a 250-mL flask at 30°C and then washed. They were transferred to 50 mL Falcon tubes and centrifuged for five minutes at 3000 rcf to pellet in an Allegra 25R centrifuge (Beckman Coulter). The supernatant was discarded, and the cells were resuspended in 25 mL deionized water. They were again centrifuged for five minutes at 3000 rcf to pellet. The supernatant was discarded, and the cells were resuspended in 1 mL of deionized water. They were then centrifuged at 20,000 rcf to pellet on a benchtop centrifuge. The supernatant was discarded, and the cells were resuspended in 600 microliters of deionized water. The pre-stress time points for all experiments were taken from this cell suspension.

To stress the cells with ethanol or hydrogen peroxide, 400 microliters of the suspension was added to 600 microliters of solution containing the appropriate stress, immediately vortexed, set at room temperature for two minutes, vortexed again, and immediately removed from the

stress. The 600 microliters of solution contained the level of the stressor such that the final 1 mL of cell suspension contained the desired concentration of ethanol or hydrogen peroxide. For example, if the cells were to experience a stress level of 20% ethanol, 200 microliters of 100% ethanol were added to 400 microliters of deionized water. As soon as 400 microliters of cells were added to this solution, the final 1 mL volume of cells would immediately encounter the 20% ethanol stress. In the case of heat stress, 400 microliters of the pre-stress cell suspension were added to 600 microliters of deionized water. This 1 mL of cell suspension was vortexed, placed in a water bath at the desired temperature for two minutes, vortex again, and immediately removed from the stress.

For ethanol and hydrogen peroxide stress, the stress was removed through dilution. The 1 mL of stressed cells was diluted tenfold into deionized water and vortexed. The post-stress time points for all experiments were taken from this final cell suspension, whether it be for recovery or plating. A control experiment was performed in which cells were subjected to the tenfold diluted concentration of the desired stress. They were in this mild stress for four hours and plated periodically on sc+glu plates. There was no loss in viability in any of the tenfold diluted concentrations of either hydrogen peroxide or ethanol stress.

For all experiments in which fraction survival was calculated, 100 microliters were removed from the pre-stress and post-stress cell suspension for serial dilutions in deionized water and then plated on sc+glu plates. The number of colonies (CFU) from the plates were counted daily until the CFU count no longer increased. The number of cells before and after stress were calculated from the CFU counts. Fraction survival was defined as the ratio of post-stress cells to pre-stress cells.

## Transcriptional profiling using RNA-seq

Diagrammatically shown in Fig 1B, there were five time points taken for each experiment, which will be referred to as growth, pre-stress, 15 minutes post-stress, 30 minutes post-stress, and 60 minutes post-stress. Yeast colonies were picked from YPD plates and grown overnight in 2 mL of sc+glu media at 30˚C until saturation. The next day, cells were back-diluted 1:200 into 25 mL of fresh, prewarmed sc+glu media in a 250-mL flask and grown for 6 hours at 30˚C. Before washing the cells in preparation for ethanol stress exposure, 1.5 mL of cells were removed and centrifuged at 20,000 rcf for 1 minute to pellet. The supernatant was removed, and the pellet was immediately frozen and stored at -80˚C. This sample is used to determine gene expression levels in the "optimal" growth conditions. The rest of the 23.5 mL of cells were then washed and stressed as described above in "Stress experiments". From the 600-microliter pre-stress cell suspension, 150 microliters were removed, pelleted, and frozen, similar to the growth time point. This sample is the pre-stress time point, used to determine gene expression levels immediately prior to stress exposure. The cells were recovered from the ethanol stress in deionized water. During this recovery period, 1.5 mL of cells were removed at 15 minutes, 30 minutes, and 60 minutes to pellet and freeze. All samples were stored at -80˚C until preparation for sequencing.

## Fitness profiling of a pooled haploid yeast deletion library

Diagrammatically shown in Fig 2A, the pooled yeast deletion library was thawed from a frozen stock and grown in 25 mL of sc+glu media in a 250-mL flask at 30˚C until mid-log phase. Samples were then washed and stressed as described above in "Stress experiments". 150 microliters of the pre-stress cell suspension was saved as the pre-stress time point sample. The post-stress cell suspension was subjected to an outgrowth period in sc+glu media at 30˚C until the cells reached early log phase, which was then saved as the post-stress time point sample. All experiments were performed with 24.5% ethanol and in triplicates.

## Laboratory experimental evolution for enhanced survival to lethal ethanol stress

Diagrammatically shown Fig 3A, yeast colonies were picked from YPD plates and grown overnight in 2 mL of sc+glu media at 30˚C until saturation. The next day, cells were back-diluted 1:200 into 25 mL of fresh, prewarmed sc+glu media in a 250-mL flask at 30˚C, and grown for six hours. The cells were then washed and stressed, and fraction survival was also calculated as described above in "Stress experiments". From the ten-fold diluted post-stress cell resuspension, 500 microliters were added to 2 mL of sc+glu media and regrown in 30˚C until saturation, thus starting the next round of evolution. Whenever cells were back-diluted 1:200 for the six-hour growth, 300 microliters of cells were frozen and stored in -80˚C to capture the cellular state at every round of evolution. All experiments continued for 10–12 rounds of evolution. Three parallel lines of laboratory evolution were performed for any given strain background and stress condition. At the end of the evolution experiment, one evolved population from each replicate line was chosen for further analysis and for sequencing.

These experiments were designed in such a way as to select for both stress defense as well as fast growth. While certain laboratory evolution studies use nutrient-limited media during the recovery period after stress to limit selection for strains that grow faster or recover from the stress faster [48], we wished to avoid accumulation of mutations that cause generic slow growth, which would be a trivial solution to lethal stress survival given the established link between growth rate and stress defense. This strategy favored mutations that increased survival without compromising growth rate or recovery time. Since slow-growing mutations will be positively selected for during stress exposure and negatively selected for during recovery, we saw non-monotonic and cyclical patterns of survival. As such, the last round of evolution was not necessarily when the evolved cells survive the best to lethal stress. Upon noticing this observation, we altered our approach for choosing the evolved population to use for sequencing and other analyses. We selected the evolved population that was highest for surviving lethal stress based on the fraction survival data from the platings, rather than just using the evolved population from the last round of evolution.

Each evolved population was tested for stress survival under the same stress used to evolve them, using the protocol described above in "Stress experiments". The population was then streaked on sc+glu plates and individual colonies were chosen to test for survival. We avoided picking the smallest colonies so as to not bias for slow growing cells. At least six colonies from each population were tested for survival. The highest surviving colony overall was used for transcriptional profiling by RNA-seq.

## Transcriptional profiling by RNA-seq

RNA was isolated using the YeaStar RNA Kit (Zymo Research). Between step 5 and 6 of the protocol, the sample was treated with DNase I (Sigma-Aldrich). An in-column DNase digestion was performed according to Appendix A of RNA Clean & Concentrator-5 (Zymo Research). From the isolated RNA, rRNA was removed using the Ribo-Zero rRNA Removal Kit (Illumina). Samples were barcoded and prepared for sequencing using the NEBNext Ultra Directional RNA Library Prep Kit for Illumina (New England Biolabs). All samples were pooled and sequenced using a NextSeq 500 sequencer (Illumina).

## Sequencing of fitness profiling experiment

DNA was isolated with the YeaStar Genomic DNA Kit (Zymo Research) and the pooled deletion library barcodes were amplified in parallel for subsequent sequencing. For a given gene

deletion, it is replaced, immediately downstream of the start codon, with an 18-nucleotide universal priming site U1 (GATGTCCACGAGGTCTCT), a unique 20-nucleotide TAG sequence, another 18-nucleotide universal priming site U2 (CGTACGCTGCAGGTCGAC), a 1537 base pair KanMX cassette that contains the *KAN* gene, a 19-nucleotide universal priming site D2 (CGAGCTCGAATTCATCGAT), a different unique 20-nucleotide TAG sequence, a 17-nucleotide universal priming site (CTACGAGACCGACACCG), and ends with a TAA stop codon, which replaces the normal stop codon for that gene. The *KAN* gene confers resistance to the antibiotic kanamycin in bacteria and the antibiotic geneticin in yeast. Since not all gene deletions had an annotated downstream TAG, only the upstream TAG was sequenced.

In order to use Illumina platform for sequencing, Illumina compatible adapter were required to be added to the TAG sequence. We decided to use the Illumina Truseq adapters. The two universal priming sites (U1 and U2) were used to add the Truseq adapters. This was performed using two rounds of PCR that resulted in a final sequence of the form AATGATAC GGCGACCACCGAGATCTACACTCTTTCCCTACACGACGCTCTTCCGATCT*N[6–11]***ATGGA TGTCCACGAGGTCTCT**N̲N̲N̲N̲N̲N̲N̲N̲N̲N̲N̲N̲N̲N̲N̲N̲N̲N̲N̲N̲*CGTACG CTGCAGGTCGA C*AGATCGGAAGAGCACACGTCTGAACTCCAGTCAC N̲N̲N̲N̲N̲N̲ATCTCGTATGCCGTCTTCT GCTTG. The sequence is flanked by 5' and 3' ends of the Illumina Truseq adapters (normal font), with the underlined sequence indicating a specific sequence for multiplexing different samples. We used Truseq adapters 2, 4, 5, and 6. The bold, bold/underlined, and bold/italic sequences represent the U1, TAG, and U2 sequences, respectively. The italic section indicates custom-made internal indices (ATCACG, TTAGGCG, ACTTGACG, GATCAGTAG, TAGCTTA‐ CAG, GGCTACGAGTG) for additional multiplexing. These internal indices were made in such a way as to minimize failure during sequencing due to overabundance of one type of nucleotide. All samples were pooled and sequenced using a NextSeq 500 sequencer (Illumina).

## Pre-processing of sequencing results

All sequencing reads were first separated based on sample number using bcl2fastq conversion software (Illumina). For RNA-seq, sequencing reads were clipped to remove Illumina adapter sequences (AGATCGGAAGAGC) using cutadapt [56]. They were then trimmed using Trimmomatic 0.33 [57] to remove end bases with a quality score below three and retain only the part of the read that has quality score above fifteen using a four-base sliding window. Reads that were less than ten base pairs as a result of this trimming were discarded. For the deletion library data, a second demultiplex step, using a custom-written script, was performed after the initial bcl2fastq to separate samples based on internal staggered indices. By searching for the U1 and U2 primer sequences, we were able to find the unique 20-nucleotide TAG sequence in each sequencing read. All reads that did not contain a TAG sequence were discarded.

## Mapping of sequencing reads

For the RNA-seq data, reads were aligned to the reference transcriptome of *S. cerevisiae* strain S288C (https://downloads.yeastgenome.org/sequence/S288C_reference/orf_dna/) using kallisto [58]. Most of our samples had greater than 90% of their reads mapped to the S288C transcriptome, with the lowest at 85.6%. The counts for each gene from the kallisto output were expressed in the normalized form, transcripts per million (TPM). TPM counts for each sample were then combined into a matrix, and genes in which all samples had a raw read count below ten were discarded from further analysis.

For the deletion library data, reads were mapped to the reference file of all TAG sequences using bowtie2, default settings [59]. All reads that mapped to multiple TAG sequences were discarded. A total of 4,563 gene deletions were represented in our samples. The total number

of TAG sequences was normalized across all samples, and all gene deletions in which the read count was less than ten was discarded.

## Clustering of RNA-seq data and functional category analysis

For clustering, all genes in which the coefficient of variation across all time points was below 0.2 were removed. The remaining genes were clustered in R using k-means, with a range of cluster sizes from 5–30. The genes and their cluster designations were used as input to iPAGE (Pathway Analysis of Gene Expression) to identify the likely pathways that are overrepresented in each of the clusters [14]. We decided to use a cluster size of ten for all downstream analyses because it provided the best balance between the number of distinct transcriptional dynamics and minimal redundancy in the pathways that were enriched within them. As can be seen from Fig 1D, one of the ten clusters (cluster 8) does not have any significant pathways. Increasing the number of clusters increases the number of empty clusters. To expand the list of relevant pathways, iPAGE analysis was also performed on each of the individual post-stress time points after being mathematically zero-transformed by the pre-stress time point. The full list of pathways was then filtered for only significant terms (p<0.001). The list of pathways was large and had many redundant terms, so it was summarized into functional categories using REVIGO [60]. In addition to k-means clustering, we also performed hierarchical clustering of both genes and time points, using the average linkage clustering method.

## Enrichment/depletion and functional category analysis

DEseq2 was used to determine gene deletions that are significantly enriched or depleted after stress survival [61]. We chose to use DESeq2 based on our data fitting a negative binomial distribution. Robinson et al., 2014 [62] demonstrated that approaches based on negative binomial models, such as DESeq, that were originally used to analyze RNA-seq data, were directly applicable to Bar-seq data. Enrichment score was determine based on the average fitness of triplicates, and significance was determined by a Benjamini-Hochberg adjusted p-value < 0.05. The resulting enrichment scores were used as input to iPAGE. The common functional categories between the yeast deletion library data and the RNA-seq data were then compared to determine if any association existed between the two datasets.

## Supporting information

**S1 Fig. Clustering of the time-course gene expression data of the wild-type strain using hierarchical clustering.** Both genes and time points were clustered, with all time points adjusted to be relative to the pre-stress time point.
(TIF)

**S2 Fig. Overrepresented motifs from condensed chromosome and spore wall assembly genes.** Using FIRE *de novo* motif discovery on condensed chromosome and spore wall assembly genes, two overrepresented motifs in the promoter region of those genes were discovered.
(TIF)

**S3 Fig. Over- and underrepresented pathways within the full range of fitness scores.** The pathways that are over- or underrepresented in ethanol stress. Overrepresented pathways are shown in red and underrepresented pathways are shown in blue.
(TIF)

**S4 Fig. Tracking survival at each round of laboratory evolution.** The fraction survival was calculated at each round of laboratory evolution for all three replicate lines of the wild-type background.
(TIF)

**S5 Fig. Clustering of the time-course gene expression data of strain JY304 using hierarchical clustering.** Both genes and time points were clustered, with all time points adjusted to be relative to the pre-stress time point.
(TIF)

**S6 Fig. Major functional categories discovered in the post-stress period at the threshold of lethality.** Categories in red are those significantly upregulated post-stress. Categories in blue are those significantly downregulated post-stress.
(TIF)

**S1 Table. All functional categories modulated during the adaptation process of haploid yeast exposed to threshold lethal ethanol stress.**
(XLS)

**S2 Table. Comparison of pathways and genes affected following mild ethanol stress versus acute lethal ethanol stress.** This table lists functional categories (top) that are affected by mild ethanol stress or individual genes (bottom) that are important in mild ethanol stress survival. The directionality of the mild ethanol stress response is compared to whether our wild-type cells increases/decreases expression of the gene/pathway and whether deletion of the gene/pathway increases/decreases fitness after acute lethal ethanol exposure. A blank indicates that the pathway or gene did not significantly change in our data.
(XLSX)

**S3 Table. All significantly differentiated functional categories between strain JY304 and the wild-type strain during exponential growth.**
(XLS)

## Acknowledgments

We thank the Tavazoie laboratory for many helpful discussions and guidance.

## Author Contributions

**Conceptualization:** Jamie Yang, Saeed Tavazoie.

**Formal analysis:** Jamie Yang.

**Funding acquisition:** Saeed Tavazoie.

**Investigation:** Jamie Yang.

**Methodology:** Jamie Yang.

**Project administration:** Saeed Tavazoie.

**Supervision:** Saeed Tavazoie.

**Validation:** Jamie Yang.

**Writing – original draft:** Jamie Yang, Saeed Tavazoie.

**Writing – review & editing:** Jamie Yang, Saeed Tavazoie.

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
