## [Decision Letter · Decision Letter 0]

7 Jul 2020

PONE-D-20-17709

Regulatory and evolutionary adaptation of yeast to acute lethal ethanol stress

PLOS ONE

Dear Dr. Tavazoie,

Thank you for submitting your manuscript to PLOS ONE. After careful consideration, we feel that it has merit but does not fully meet PLOS ONE’s publication criteria as it currently stands. Therefore, we invite you to submit a revised version of the manuscript that addresses the points raised during the review process.

We look forward to receiving your revised manuscript.

Kind regards,

Alvaro Galli

Academic Editor

PLOS ONE

Journal Requirements:

2.Thank you for stating the following in the Acknowledgments Section of your manuscript:

'JY was supported by the NIH MSTP MD-PhD training program. ST was supported by grants from the NIH (R01-AI077562 and R01-HG009065).'

 'The funders had no role in study design, data collection and analysis, decision to publish, or preparation of the manuscript.'

Reviewers' comments:

Reviewer's Responses to Questions

**Comments to the Author**

1. Is the manuscript technically sound, and do the data support the conclusions?

Reviewer #1: Yes

Reviewer #2: Yes

Reviewer #3: Yes

2. Has the statistical analysis been performed appropriately and rigorously? 

Reviewer #1: Yes

Reviewer #2: I Don't Know

Reviewer #3: Yes

3. Have the authors made all data underlying the findings in their manuscript fully available?

Reviewer #1: No

Reviewer #2: Yes

Reviewer #3: No

4. Is the manuscript presented in an intelligible fashion and written in standard English?

Reviewer #1: Yes

Reviewer #2: Yes

Reviewer #3: Yes

5. Review Comments to the Author

Reviewer #1: The authors investigated multiple aspects of yeast response and adaptation to severe ethanol stress. They specifically focus on survival after a pulse of highly lethal stress, rather than response to mild stress that frequently studies. The authors employed a very clever experimental setup that allowed them to study this phenomenon and utilize a wide variety of methodologies to investigate the underlying mechanisms, including RNA-seq for expression profiling, a genetic screen with a barcoded deletion library, and lab evolution experiments. The results described match observations of previous studies and provide new insights that will be interesting to many in the field. The key result the author described, the ethanol evolved strains reprogramming their pre-stress transcriptional program so it matches the post-stress transcriptional program in the ancestor strain, is interesting and has implications far beyond the S. cerevisiae model system.

Overall, this work’s approach for observing the response to lethal ethanol exposure is both unique and complementary to the current understanding of sub-lethal stress responses. The main limitation of the work is the lack of biological replicates for some of the high-throughput experiments which limits the capacity to generalize the interesting observations the author made. Additionally, the methodology used for deriving biological insight from gene expression profiling are non-intuitive and may be suboptimal for this dataset. An alternative and more common approach should be considered instead. To summarize, the manuscript is overall written well and describes interesting findings in the yeast stress response that will be interesting to many researchers. I recommend only minor revisions before its publication.

Major comments

1. The experimental setup of brief stress exposure is clever and uncommon. At least two reasons can explain why only some cells survive while other don’t. Either (1) cells survived since they were able to mount a post-stress response that is essential for survival (as the propose in the abstract) or (2) that surviving cells expressed genes essential for survival pre-stress (as they mention in the introduction). The authors conclude that the post-response transcriptional response is adaptive (increases fitness) since they see a negative correlation between transcriptional change and fitness of a strain deleted for the same gene (figure 2). I am not convinced that this observation is sufficient to draw this strong conclusion. It could be that these genes are essential for ethanol survival since their basal expression prior to the stress is key for survival (and their transcriptional changes post stress is inconsequential but still correlated with survival). This conclusion can only be supported with additional experiments that will test if stress survival is diminished or is left unchanged if transcription post-stress is inhibited (of specific genes or gene sets of interest). Indeed, results from the evolved strain support the notion that the transcriptional program prior to the stress can increase stress survival. The question of whether the post-stress response is adaptive is a truly hard one and could therefore be left unanswered in the scope of this paper. However, I suggest it would be discussed in more detail in the discussion section.

2. A significant part of the pathway enrichment analysis from the RNA-seq dataset is built on functional enrichment (iPAGE) of individual genes sets found by k-means clustering (with 10 clusters). The choice of 10-cluster seems arbitrary (despite the short discussion in the methods section). In fact, many of these clusters seem highly correlated in the overall transcriptional response and therefore the functional enrichment done on individual clusters may not capture the underlying biology correctly. I suggest that the authors also provide a more straightforward analysis of the RNAseq dataset by performing hierarchical clustering. They can then extract the clusters from this analysis (for downstream iPAGE work). Accompanying this analysis with a figure showing the heatmap and dendrogram will allow readers to get an impartial representation of transcriptional changes at a single gene resolution. This analysis can be added to figures 2 and 4 as complementary subplots or it could be used to replace 1D and 4B.

3. Overall the figures would benefit from more specific labels and titles, and consistent scales. For example, figure 3 displays survival fraction in three different ways (horizontal lines and dots for clones in 3B, mean and errorbars in 3C, no errorbar in 3D, and bars with errorbars in 3E and 3F). Figure legends are also typically too short and sometimes omit key information (for example, what do the errorbars in figure 3E and 3F represent. It would also be very helpful to mark strain JY304 on its corresponding point in figure 3B and clearly indicate in the figure labels of 3C-F that the measurements are for evolved JY304 (rather than just “evolved”).

Minor comments:

1. On page 6 top paragraph, it states transcription was monitored for 4 hours but data shared only goes to 1 hour.

2. There are multiple instances that give relative amounts instead of actual numbers. For example: 85% of LOF genes exhibit anti correlation (pg 12), ESR has 2x downregulated genes and this work showed 5x downregulated genes from lethal exposure. To show relativity is scalable (pg 18), ‘Many’ early stage meiosis genes poses the URS1 motif (pg 18)

3. On page 10 the author provides a list of references (36-41). It is unclear what information is taken from these references. Is it the gene pathway affiliation?

4. Are the qualifications for significantly differentiated genes in previous ESR works and this work levels the same? If not, twice as many downregulated than upregulated genes in ESR as to 5x as many down regulated in these findings is not comparable.

5. The emphasis on these findings’ impact on industry is unclear since the acute stress used for this study is likely irrelevant for industrial ethanol production. Comments on this connection (in both the abstract and discussion) should either be further explained and supported or can be omitted altogether (the work is interesting and valuable on its own).

Reviewer #2: In this manuscript Yang and Tavazoie explore the ability of yeast to cope with acute ethanol stress. The key components of the project are:

1) Identify genes and categories of genes that change expression in response to a short (2-minute) ethanol stress that is sufficient to kill some percentage of the cells and characterize their response.

2) Identify genes that play important roles in ethanol survival using a barseq approach of a deletion library and compare this list of genes to the genes with changed expression after stress.

3) Evolve strains to cope with acute ethanol stress. Based on the best surviving strain, determine whether there is a tradeoff between growth rate and ability to survive. Determine whether there are pre- and post-stress transcriptional changes in the evolved strains compared to wild-type.

4) Determine whether the evolved response is a general stress response or is specific to ethanol.

This is a largely descriptive paper in that although it identifies genes and gene categories involved in these responses and changes, it doesn't really dig into the mechanistic basis of them. The experiments appear to be carefully and consistently done. The identification of gene categories that are enriched or depleted in the various experiments use tools developed in the Tavazoie lab.

There are some parts of the paper that could use clarification or elaboration.

- On the top of page 6 it says that monitoring the transcriptional response proceeding for four hours after the stress, but the post-stress RNA-seq timepoints are only 15, 30, and 60 minutes.

- The main comparison of pre- and post-stress transcriptional response is done at 20% ethanol when survival looks to be near 100%. On what basis shouuld we assume that the transcriptional response after 20% ethanol stress is the same as one after 23% or 24.5% ethanol stress, even for cells that don't die? This may in fact be true, and it makes sense, as the authors say, to avoid confounding the stress response with transcriptional changes due to death, but it isn't obvious that these transcriptional responses should necessarily be the same as the ones for cells that survive 24.5% ethanol (like the barseq and experimental evolution experiments)

- The word "correlation" is used several times in the text, but actual correlations aren't run (and would be inappropriate for the data in Fig. 2C since this is a decidedly non-linear pattern). How about the more general term "association"?

- Is DeSeq2 appropriate for barseq analysis? It wasn't designed for this and so using it in this way needs some justification.

- Was DeSeq2 also used for the expression analysis? The statistical (pre-clustering) analysis of RNA-seq data wasn't described so far as I can see unless this is dealth with in the downstream programs like iPAGE.

- As the authors say, the lack of a tradeoff between growth and survival is unexpected and contrary to previous findings. If there is no tradeoff then why wouldn't yeast be in this hyper-resistant pre-stress state all the time? After all, it conveys general stress protection. While I don't think the authors need to solve this mystery in this paper, it probably deserves some more discussion. Do the identities of the genes and pathways involved give any clue as to where a tradeoff might lie? Could the authors do one more experiment where they test growth rates in minimal media? Here's a (completely) speculative hypothesis: perhaps the evolved pre-stress transcriptional state renders the cells too sensitive to stress. Under the YPD growth like the authors used, things are good and the stress response isn't triggered. But under minimal media while the wild-type strain doesn't grow as fast but still grows, the evolved strain over compensates and triggers a stress response where there really isn't one. This is (as I said) completely speculative, but perhaps a few easy growth experiments in different conditions might be able to locate where a tradeoff is or further support the surprising finding that there is none.

- The evolution results are all based on a single strain. But it looks from the methods that other strains were genome sequenced (but not transriptionally profiled?) Also, if the evolved strains were genome-sequenced, was there anything informative there? It just seems like a loose end. How unusual is JY304? Do the other evolved strains also show this lack of growth/survival tradeoff? Was JY304 the one population on page 26 that fit the initial criteria?

- Where it is possible, the authors should post the code/scripts they used to run the analyses and process the raw data.

- How dependent are the results and conclusions on the number of clusters chosen for the cluster analyses? Do results change if you go to 6 or 15 instead of 10?

Reviewer #3: Summary of the research

In the manuscript “Regulatory and evolutionary adaptation of yeast to acute lethal ethanol stress”, the authors investigated the relatively untouched question of how cells survive acute stress. Most studies of stress responses in yeast (and potentially other species) were focused on uncovering the stereotypical gene expression change, which requires the experimenters to subject the cells to a sufficiently long period of stress to allow gene expression changes to reach or get close to the maximal level. In this study, however, the authors asked a different question, which is what genes are involved in the cells’ ability to survive a severe but brief exposure to the stress, specifically, they examined ethanol stress at a concentration between 19-26%.

The authors made a number of novel and significant findings. First, using transcriptome profiling by RNA-seq post-stress, they uncovered gene functional groups that are significantly up- or down-regulated. I especially appreciate the authors’ further testing of whether the gene expression changes were adaptive (helping the cells to survive and recover from the stress-induced damages) or maladaptive (as a result of and potentially cause for further damages). This is achieved by subjecting the non-essential gene deletion library in the yeast to the same stress treatment, and letting the surviving cells go through a round of outgrowth meant to remove the slow-growth mutants that would have survived the stress but are not of interest. The result revealed that the majority of the genes with expression changes are potentially adaptive, as their corresponding deletion mutants exhibit consistent fitness effects (genes down-regulated after the acute stress exhibited higher fitness when deleted and vice versa).

Not content with what they learned, they decided to subject wild-type yeast cells to an experimental evolution scheme, where the cells were exposed to an alternating environment of acute ethanol stress followed by an outgrowth in the rich media. Because the level of acute stress is such that at the beginning of the experiment, only 1% of the cells will survive, there is probably more room to improve fitness by increasing the survival following the ethanol stress than optimizing the growth rate in the rich media (which does happen, but at a small magnitude. see work from Desai lab and others). They carried out three technical replicates and chose to focus on the most successful one, which improved its survival by ~35% (there are quite a bit of heterogeneity among the three replicates in terms of their improvement in survival rates, suggesting that the genomic targets for adaptive mutations may be relatively small, and the order of acquiring those mutations may also matter).

To identify genes whose transcriptional changes in the evolved strain may underlie its higher survival, the authors compared both the post-stress and pre-stress expression states between the evolved and the parent strains. For the post-stress expression changes, they found that the evolved strain shared 6/10 functional categories of significantly induced/repressed genes compared with the parent strain, but the magnitude of changes are smaller, and there is not a significant “recovery” (regression to the pre-stress state) as the parent strain does. In terms of the pre-stress states, they revealed two significant findings. First, the most significantly differentially expressed genes mostly have unknown functions or belong to transposable elements. It is unclear whether these genes relate to the better survival trait of the evolved strain. If they do, it would suggest that our knowledge about the most well-studied eukaryotic genome is still grossly lacking! Second, their pathway analysis among the DGEs between the evolved and the parent strains pre-stress revealed many of the same categories of genes that are induced or repressed in the parent strain following the ethanol exposure, suggesting that one of the potential mechanisms behind the evolved strain’s higher survival is by making some of the transcriptional changes induced by the stress a constitutive state of the cell. It is important to note that although the authors showed that the mutant doesn’t suffer any growth disadvantage during the log phase in rich media, there could be hidden costs that are only revealed when the cells are subject to more complex environments, such as rapid adjustment to different nutrient source or growth under limited nutrients. Also, I don’t agree with the authors in their claim that this decoupling between growth and stress resistance is unexpected, because the selection scheme they implemented essentially requires the mutants to fulfill those two criteria. However, that doesn’t diminish the significance of their finding that these two factors may be separable (although, see above comments on other environments).

Criticism

Overall the experiments were very well-designed and possible explanations were considered and tested, which I very much appreciate. I would have loved to know, with more direct perturbation experiments and phenotypic assays, about how some of the most significantly induced or repressed genes contribute to the survival of the cells, following their RNA-seq analysis. Another place where I would love to know more about is the cellular heterogeneity among the population of the evolved strain -- was its better survival as a population the result of a uniform improvement of all cells, or was it due to the evolution of a bet-hedging strategy, where a small sub-population of cells were “set aside” as slow-growers and can better weather the stress. Experiments done in E. coli (e.g. “persisters”, from Stanis Leibler lab) has shown that the persistence phenomenon exists and is evolvable via genetic changes. I do believe, however, the work presented in this manuscript is sufficient for publication. The above requests were my personal curiosity and I hope/believe that they are on the authors’ todo list as well. Below, however, I list a number of suggestions and raise some concerns, in the hope that they help the authors further improve the manuscript.

Main concern:

The authors emphasized the distinction between the acute stress scheme they used in multiple places. I have two problems with respect to this. First, while it is easy to understand how the pre-stress cellular state could be important for the acute stress scheme, it’s less clear to me what they mean by “long-term expression change following the stress”. In particular, how is that different from the “canonical” ethanol stress response that the cells mount following a less acute stress scheme, as studied by the others? Second, and directly related to the above, I would love to see a direct comparison of the genes identified via the RNA-seq and deletion library screening in the acute stress with the known gene expression changes and GO categories following a more mild stress challenge, as revealed in other studies. The authors did mention one of the distinctions -- among the ESR genes, the downregulated genes show more similarity while the up-regulated ESR genes are absent among those responding to the acute stress.

Minor concerns / suggestions:

1. The introduction could be more tightly connected to the main questions and findings of the paper. For example, the authors spent a paragraph talking about the ESR. It is not clear, however, what is the relationship between the ESR and their finding until I read the Discussions. If each piece of background information can have a clear connection to the main points of the paper, it will make the introduction less arbitrary and more informative.

2. The description of the gene pathways enriched among the DGE following acute ethanol stress feel like long and diffused. This is probably a general challenge to genome-wide characterization. I wonder if the authors can attach the most important categories to a cartoon of the cell. By connecting gene categories to particular organelles of the cell or a specific biological process, e.g. DNA-repair, meiosis, bud formation etc., it will help the readers absorb and appreciate the information.

3. I would like to get the authors’ interpretation of the apparently high levels of heterogeneity in the improvement in fitness among the three replicate populations in their experimental evolution.

4. What are the major damages of high concentrations of ethanol to the cells? Can we tie the DE genes’ function to those damages?

5. In Figure 4B, it is not clear to me whether the ten categories in the mutant vs the parent strain match each other. Another way of asking this question is, is the heatmap for the parent strain the same as in Figure 1 but just reordered or are they matched to the 10 categories identified in the evolved strain? A clarification in the legend would have been helpful.

6. PLOS authors have the option to publish the peer review history of their article (what does this mean?). If published, this will include your full peer review and any attached files.

Reviewer #1: **Yes: **Amir Mitchell and Brittany Rosener

Reviewer #2: No

Reviewer #3: **Yes: **Bin He

---

## [Author Response · Author response to Decision Letter 0]

24 Aug 2020

We thank the reviewers for their insightful and extremely helpful comments. We have addressed all their concerns below and have modified the manuscript to reflect the changes/additions. The reviewer comments are in regular text. Our responses are underlined. Changes made to the manuscript are presented as italic.

Reviewer 1:

Major comments:

1. “The experimental setup of brief stress exposure is clever and uncommon. At least two reasons can explain why only some cells survive while others don’t. Either (1) cells survived since they were able to mount a post-stress response that is essential for survival (as the propose in the abstract) or (2) that surviving cells expressed genes essential for survival pre-stress (as they mention in the introduction). The authors conclude that the post-response transcriptional response is adaptive (increases fitness) since they see a negative correlation between transcriptional change and fitness of a strain deleted for the same gene (figure 2). I am not convinced that this observation is sufficient to draw this strong conclusion. It could be that these genes are essential for ethanol survival since their basal expression prior to the stress is key for survival (and their transcriptional changes post stress is inconsequential but still correlated with survival). This conclusion can only be supported with additional experiments that will test if stress survival is diminished or is left unchanged if transcription post-stress is inhibited (of specific genes or gene sets of interest). Indeed, results from the evolved strain support the notion that the transcriptional program prior to the stress can increase stress survival. The question of whether the post-stress response is adaptive is a truly hard one and could therefore be left unanswered in the scope of this paper. However, I suggest it would be discussed in more detail in the discussion section.”

Response: This is an important point raised by the reviewer. Indeed, the transcriptional change post-stress is suggestive of a fitness benefit. However, in order for us to conclusively demonstrate this, additional experiments need to be performed to precisely test this hypothesis for every gene. We have therefore softened this interpretation in the main text and have added the following to the discussion section:

“The question of whether the post-stress transcriptional response in wild-type cells is adaptive is a difficult one to conclusively answer. We believe that it does confer a fitness benefit for a couple of reasons. First, our pooled yeast deletion library experiments showed a significant negative association between transcriptional change and fitness, as shown in Fig 2C-D. Second, as shown in Table 1, the post-stress upregulation of lipid biosynthesis, ATPases, lipid transport, and vacuolar functions, all of which are normally disrupted by ethanol, suggests that the cells have mounted a response to counteract the damaging effects of ethanol. However, in order for us to conclusively demonstrate that our post-stress cells are adaptive, additional experiments would need to be performed to test individual genes of interest.”

2. “A significant part of the pathway enrichment analysis from the RNA-seq dataset is built on functional enrichment (iPAGE) of individual genes sets found by k-means clustering (with 10 clusters). The choice of 10-cluster seems arbitrary (despite the short discussion in the methods section). In fact, many of these clusters seem highly correlated in the overall transcriptional response and therefore the functional enrichment done on individual clusters may not capture the underlying biology correctly. I suggest that the authors also provide a more straightforward analysis of the RNAseq dataset by performing hierarchical clustering. They can then extract the clusters from this analysis (for downstream iPAGE work). Accompanying this analysis with a figure showing the heatmap and dendrogram will allow readers to get an impartial representation of transcriptional changes at a single gene resolution. This analysis can be added to figures 2 and 4 as complementary subplots or it could be used to replace 1D and 4B.”

Response: We had originally performed a range of partitional and hierarchical clustering analyses on these expression data (including varying the number of clusters) and had concluded that the current presentation of 10 clusters best captures the broad dynamics observed. However, we agree with the reviewer that adding a hierarchical clustering of these data would be a very useful representation. We have, thus, performed these analyses and have provided them in the Supplementary Figures S1 Fig and S5 Fig for both the parental and evolved strains. Additionally, we have added additional text describing our observation from the hierarchical clustering.

3. “Overall the figures would benefit from more specific labels and titles, and consistent scales. For example, figure 3 displays survival fraction in three different ways (horizontal lines and dots for clones in 3B, mean and errorbars in 3C, no errorbar in 3D, and bars with errorbars in 3E and 3F). Figure legends are also typically too short and sometimes omit key information (for example, what do the errorbars in figure 3E and 3F represent. It would also be very helpful to mark strain JY304 on its corresponding point in figure 3B and clearly indicate in the figure labels of 3C-F that the measurements are for evolved JY304 (rather than just “evolved”).”

Response: We thank the reviewer for these helpful suggestions. The figures and figure legends have been updated to reflect these.

Minor comments:

1. “On page 6 top paragraph, it states transcription was monitored for 4 hours but data shared only goes to 1 hour.”

Response: We thank the reviewer for pointing out this error. We have corrected the text from “four hours” to “one hour”.

2. “There are multiple instances that give relative amounts instead of actual numbers. For example: 85% of LOF genes exhibit anti correlation (pg 12), ESR has 2x downregulated genes and this work showed 5x downregulated genes from lethal exposure. To show relativity is scalable (pg 18), ‘Many’ early stage meiosis genes poses the URS1 motif (pg 18).”

Response: We thank the reviewer for this suggestion. We have added absolute numbers to relevant places in the paper, including the ones mentioned on pages 12 and 18.

3. “On page 10 the author provides a list of references (36-41). It is unclear what information is taken from these references. Is it the gene pathway affiliation?”

Response: Yes, these are references to the genes/pathways. We have now split these citations to more clearly link them with their corresponding genes/pathways.

4. “Are the qualifications for significantly differentiated genes in previous ESR works and this work levels the same? If not, twice as many downregulated than upregulated genes in ESR as to 5x as many down regulated in these findings is not comparable.”

Response: Yes, in references 4 and 5, the original ESR papers, both groups used two-fold gene expression cutoffs for downregulated and upregulated genes, so we used the same criteria for our paper.

5. “The emphasis on these findings’ impact on industry is unclear since the acute stress used for this study is likely irrelevant for industrial ethanol production. Comments on this connection (in both the abstract and discussion) should either be further explained and supported or can be omitted altogether (the work is interesting and valuable on its own).”

Response: We appreciate this point raised by the reviewer and have omitted the findings’ impact on industry in both the abstract and discussion.

Reviewer 2:

1. “On the top of page 6 it says that monitoring the transcriptional response proceeding for four hours after the stress, but the post-stress RNA-seq timepoints are only 15, 30, and 60 minutes.”

Response: We thank the reviewer for pointing out this error. We have corrected the text from “four hours” to “one hour”.

2. “The main comparison of pre- and post-stress transcriptional response is done at 20% ethanol when survival looks to be near 100%. On what basis should we assume that the transcriptional response after 20% ethanol stress is the same as one after 23% or 24.5% ethanol stress, even for cells that don't die? This may in fact be true, and it makes sense, as the authors say, to avoid confounding the stress response with transcriptional changes due to death, but it isn't obvious that these transcriptional responses should necessarily be the same as the ones for cells that survive 24.5% ethanol (like the barseq and experimental evolution experiments).”

Response: We appreciate the reviewer’s perspective here, and we agree that there may be some differences in the transcriptional responses between 20% and 23-24% ethanol stress. However, we expect the core responses to be very similar. We have added the following to the discussion in order to point out this assumption:

“When comparing the response to ethanol at 20% (transcriptional profiling of wild-type cells) versus the response at 24.5% ethanol (fitness profiling of deletion library), we are making the assumption that the core response to ethanol stress is similar across the regime spanning 20-24.5% ethanol. As shown by the top section of S2 Table, the pathways affected by mild ethanol stress versus acute lethal ethanol stress is very much in agreement, and we believe this core response extends into the realm of lethality. For example, downregulation of ribosomal proteins and translation are universal responses at various ethanol concentrations below, at, and above 20% ethanol. We do acknowledge, however, that there may be some differences in the transcriptional responses between 20% and 24.5% ethanol. Our analysis therefore attempts to capture the most significant global similarities and differences.”

3. “The word "correlation" is used several times in the text, but actual correlations aren't run (and would be inappropriate for the data in Fig. 2C since this is a decidedly non-linear pattern). How about the more general term "association"?”

Response: We agree and have changed “correlation” to “association” or “relationship” in instances where an actual correlation was not determined.

4. “Is DeSeq2 appropriate for barseq analysis? It wasn't designed for this and so using it in this way needs some justification.”

Response: We added the following justification in our methods as to why we chose DESeq2:

“We chose to use DESeq2 based on our data fitting a negative binomial distribution. Robinson et al., 2014 (reference 62) demonstrated that approaches based on negative binomial models, such as DESeq, that were originally used to analyze RNA-seq data, were directly applicable to Bar-seq data.”

5. “Was DeSeq2 also used for the expression analysis? The statistical (pre-clustering) analysis of RNA-seq data wasn't described so far as I can see unless this is dealt with in the downstream programs like iPAGE.”

Response: DESeq2 was not used for the expression analysis. After mapping the RNA-seq reads, we converted the counts for each gene into transcripts per million, and removed any genes in which all time points had a count less than 10 (Methods section “Mapping of sequencing reads”). Before clustering, we removed all genes in which the coefficient of variation was below 0.2 (Methods section “Clustering of RNA-seq data and functional category analysis”). We did not filter the results further based on any statistical analysis because we wanted the input for clustering and iPAGE to be as unbiased as possible.

6. “As the authors say, the lack of a tradeoff between growth and survival is unexpected and contrary to previous findings. If there is no tradeoff then why wouldn't yeast be in this hyper-resistant pre-stress state all the time? After all, it conveys general stress protection. While I don't think the authors need to solve this mystery in this paper, it probably deserves some more discussion. Do the identities of the genes and pathways involved give any clue as to where a tradeoff might lie? Could the authors do one more experiment where they test growth rates in minimal media? Here's a (completely) speculative hypothesis: perhaps the evolved pre-stress transcriptional state renders the cells too sensitive to stress. Under the YPD growth like the authors used, things are good and the stress response isn't triggered. But under minimal media while the wild-type strain doesn't grow as fast but still grows, the evolved strain over compensates and triggers a stress response where there really isn't one. This is (as I said) completely speculative, but perhaps a few easy growth experiments in different conditions might be able to locate where a tradeoff is or further support the surprising finding that there is none.”

Response: Although this lack of tradeoff between growth and survival is unexpected and contrary to previous findings, there is definitely a possibility that certain environments exist in which our evolved strains would perform worse than wild-type strains. Just as the environmental stress response conveys general stress protection, it does not explain behavior in some laboratory stresses, such as various nutrient deprivations. It is certainly possible that our evolved strain performs sub-optimally in those environments as well. Additionally, this lack of tradeoff applies to acute lethal stresses as described in our manuscript, but we have no idea how the two strains compare under stresses of long durations. How the evolved cells hold up in numerous other laboratory conditions and environments is a good subject for future studies. We have expanded our discussion section to include this point as shown below:

“While the survival advantage of strain JY304 was not accompanied by any significant reduction in the bulk growth rate of the population, there is a possibility that certain environments exist in which strain JY304 would exhibit lower fitness than the wild-type strain. Just like the ESR conveys general stress protection, it does not explain behavior in some laboratory stresses, such as various nutrient deprivations (6). It is certainly possible that strain JY304 does not perform optimally in those environments as well. Additionally, this lack of tradeoff between growth rate and survival applies to acute lethal stresses, but we have not determined how the two strains compare under stresses of long durations. How our evolved strain holds up in numerous other laboratory conditions and environments would be an important subject for future studies.”

7. “The evolution results are all based on a single strain. But it looks from the methods that other strains were genome sequenced (but not transcriptionally profiled?) Also, if the evolved strains were genome-sequenced, was there anything informative there? It just seems like a loose end. How unusual is JY304? Do the other evolved strains also show this lack of growth/survival tradeoff? Was JY304 the one population on page 26 that fit the initial criteria?”

Response: We had preliminary whole-genome sequencing results from some of the evolved strains but did not find any significant recurring mutations. We have, thus, removed this section of the methods since these results were not mentioned anywhere else in our paper. In addition, since our goal was to compare the evolved strain with the highest survival to the wild-type strain, we did not transcriptionally profile the other strains that showed inferior survival.

8. “Where it is possible, the authors should post the code/scripts they used to run the analyses and process the raw data.”

Response: We have deposited the README file describing our general steps has the following DOI: https://www.protocols.io/view/yang-tavazoie-sequencing-readme-bjt5knq6. The relevant code/scripts described in the README file are located at https://github.com/yang-jamie/SCRIPTS.

9. “How dependent are the results and conclusions on the number of clusters chosen for the cluster analyses? Do results change if you go to 6 or 15 instead of 10?”

Response: We had performed the clustering using a range of clusters from 5-30. The ten-cluster partition provided the best balance between the number of distinct transcriptional dynamics and minimal redundancy in the pathways that were enriched within them. We have revised the Methods section “Clustering of RNA-seq data and functional category analysis” to clarify this point.

Reviewer 3:

Main comments:

1. “The authors emphasized the distinction between the acute stress scheme they used in multiple places. I have two problems with respect to this. First, while it is easy to understand how the pre-stress cellular state could be important for the acute stress scheme, it’s less clear to me what they mean by “long-term expression change following the stress”. In particular, how is that different from the “canonical” ethanol stress response that the cells mount following a less acute stress scheme, as studied by the others?”

Response: Since the “canonical” ethanol stress response was determined at ethanol concentrations well below the threshold of lethality, we cannot assume that they are necessarily similar to the response at the threshold of lethality. Our aim therefore was to determine the similarities and differences that do exist between them.

2. “Second, and directly related to the above, I would love to see a direct comparison of the genes identified via the RNA-seq and deletion library screening in the acute stress with the known gene expression changes and GO categories following a more mild stress challenge, as revealed in other studies. The authors did mention one of the distinctions -- among the ESR genes, the downregulated genes show more similarity while the up-regulated ESR genes are absent among those responding to the acute stress.”

Response: We thank the reviewer for this suggestion. We have now added a supplemental table (S2 Table) and additional text to reflect this comparison. See below:

“S2 Table summarizes the pathways and genes that are affected during mild ethanol stress exposure and compares the directionality of their post-stress responses to the response of our data at or above the threshold of lethality. From the top section of the table, it is clear that all pathways affected by mild stress are affected in the same direction by both our transcriptional profiling and deletion library data. On the gene level, however, only 20 out of 50 genes trend in the same direction from our RNA-seq data, and only 8 out of 50 genes agree from our deletion library data. Individual genes do not overlap significantly between mild and lethal ethanol stress. In fact, even between 8% and 11% ethanol, different genes were associated with ethanol tolerance. However, on the pathway level, there is significant agreement between the responses to mild and lethal ethanol stress.”

Minor comments:

1. “The introduction could be more tightly connected to the main questions and findings of the paper. For example, the authors spent a paragraph talking about the ESR. It is not clear, however, what is the relationship between the ESR and their finding until I read the Discussions. If each piece of background information can have a clear connection to the main points of the paper, it will make the introduction less arbitrary and more informative.”

Response: We thank the reviewer for this suggestion. The discussion of ESR was included because of its dominance in the literature of yeast stress response and our subsequent comparisons throughout the paper. We have now slightly modified the introduction to make this link more tangible.

2. “The description of the gene pathways enriched among the DGE following acute ethanol stress feel like long and diffused. This is probably a general challenge to genome-wide characterization. I wonder if the authors can attach the most important categories to a cartoon of the cell. By connecting gene categories to particular organelles of the cell or a specific biological process, e.g. DNA-repair, meiosis, bud formation etc., it will help the readers absorb and appreciate the information.”

Response: We thank the reviewer for this helpful suggestion. We have now added a cartoon figure, mentioned in the discussion and designated as supplementary figure S6 Fig, as shown below:

3. “I would like to get the authors’ interpretation of the apparently high levels of heterogeneity in the improvement in fitness among the three replicate populations in their experimental evolution.”

Response: As described in our methods section “Laboratory experimental evolution for enhanced survival to lethal ethanol stress”, our experiments were designed in such a way as to select for both stress defense as well as fast growth. Since slow-growing beneficial mutations will be positively selected for during stress exposure and negatively selected for during recovery, it may explain the non-monotonic and cyclical pattern of survival throughout the laboratory evolution process. Especially at the earlier cycles of the evolution, since only 1% of the yeast cells survive, there is a large bottleneck effect after each stress event, leading to high levels of heterogeneity in each replicate population.

4. “What are the major damages of high concentrations of ethanol to the cells? Can we tie the DE genes’ function to those damages?”

Response: We thank the reviewer for this suggestion. The dominant process by which yeast cells die from high concentrations of ethanol is increased membrane fluidity and subsequent decrease in membrane integrity. Other damages caused by ethanol include inhibition of H+-ATPase activity, inhibition of transport processes, and disruption of vacuole morphology. Our analyses suggest that our post-stress cells have altered their gene expression levels to counteract these damaging effects caused by ethanol. We have added a new table (now Table 1) to tie in the DE genes’ functions to those damages.

5. “In Figure 4B, it is not clear to me whether the ten categories in the mutant vs the parent strain match each other. Another way of asking this question is, is the heatmap for the parent strain the same as in Figure 1 but just reordered or are they matched to the 10 categories identified in the evolved strain? A clarification in the legend would have been helpful.”

Response: We thank the reviewer for pointing out this confusion. The ten categories in the evolved and parent strain are not the same. The parent strain heatmap is unaltered from Fig 1D. We have now added this clarification to the figure caption.

---

## [Decision Letter · Decision Letter 1]

9 Sep 2020

Regulatory and evolutionary adaptation of yeast to acute lethal ethanol stress

PONE-D-20-17709R1

Dear Dr. Tavazoie,

We’re pleased to inform you that your manuscript has been judged scientifically suitable for publication and will be formally accepted for publication once it meets all outstanding technical requirements.

Kind regards,

Alvaro Galli

Academic Editor

PLOS ONE

Additional Editor Comments (optional):

Reviewers' comments:

Reviewer's Responses to Questions

**Comments to the Author**

1. If the authors have adequately addressed your comments raised in a previous round of review and you feel that this manuscript is now acceptable for publication, you may indicate that here to bypass the “Comments to the Author” section, enter your conflict of interest statement in the “Confidential to Editor” section, and submit your "Accept" recommendation.

Reviewer #2: All comments have been addressed

Reviewer #3: All comments have been addressed

2. Is the manuscript technically sound, and do the data support the conclusions?

Reviewer #2: Yes

Reviewer #3: Yes

3. Has the statistical analysis been performed appropriately and rigorously? 

Reviewer #2: Yes

Reviewer #3: Yes

4. Have the authors made all data underlying the findings in their manuscript fully available?

Reviewer #2: Yes

Reviewer #3: No

5. Is the manuscript presented in an intelligible fashion and written in standard English?

Reviewer #2: Yes

Reviewer #3: Yes

6. Review Comments to the Author

Reviewer #2: The authors have addressed my concerns. I hope they continue to explore what they've discovered in future studies - I look forward to learning what they find.

Two more minor word choice comments:

strains vs. strain in the sentences starting "This experimental evolution paradigm..."

In the discussion: "When comparing...versus lethal ethanol stress _are_ very much..."

Reviewer #3: (No Response)

7. PLOS authors have the option to publish the peer review history of their article (what does this mean?). If published, this will include your full peer review and any attached files.

Reviewer #2: No

Reviewer #3: **Yes: **Bin He

---

## [Editor Report · Acceptance letter]

22 Sep 2020

PONE-D-20-17709R1 

Regulatory and evolutionary adaptation of yeast to acute lethal ethanol stress 

Dear Dr. Tavazoie:

I'm pleased to inform you that your manuscript has been deemed suitable for publication in PLOS ONE. Congratulations! Your manuscript is now with our production department. 

Kind regards, 

on behalf of

Dr. Alvaro Galli 

Academic Editor

PLOS ONE